

# Monitoring small reservoirs storage from satellite remote sensing in inaccessible areas

Nicolas Avisse[1], Amaury Tilmant[1], Marc François Müller[2], and Hua Zhang[3]

[1]Department of Civil Engineering and Water Engineering, Université Laval, Québec, QC G1V 0A6, Canada
[2]Department of Civil & Environmental Engineering & Earth Science, University of Notre Dame, Notre Dame, IN 46556, USA
[3]Department of Engineering, School of Engineering and Computing Sciences, Texas A&M University - Corpus Christi, Corpus Christi, TX 78412, USA

*Correspondence to:* Nicolas Avisse (nicolas.avisse@gmail.com)

**Abstract.** In river basins with water storage facilities, the availability of regularly-updated information on reservoir level and capacity is of paramount importance for the effective management of those systems. Yet, for the vast majority of reservoirs around the world, storage levels are either not measured or not readily available due to financial, political or legal considerations. This paper proposes a novel approach using Landsat imagery and Digital Elevation Models (DEM) to retrieve information on storage variations in inaccessible regions. Unlike existing approaches, the method does not require any *in situ* measurement and is appropriate to monitor small, and often undocumented, irrigation reservoirs. It consists of three recovery steps: (i) a 2D dynamic classification of Landsat spectral bands information to quantify the surface area of water, (ii) a statistical correction of DEM data to characterize the topography of each reservoir and (iii) a 3D reconstruction algorithm to correct for clouds and Landsat 7 Scan Line Corrector failure. The method is applied to quantify reservoir storage in the Yarmouk basin in Southern Syria, where ground monitoring is impeded by the ongoing civil war. It is validated against available *in situ* measurements in neighboring Jordanian reservoirs. Coefficients of determination range from 0.69 to 0.84, and the average relative error from 3 % to 35 % for storage estimations on six Jordanian reservoirs with maximal water surface areas ranging from 0.59 km$^2$ to 3.79 km$^2$.

## 1 Introduction

Reservoirs are essential for the development and management of a river basin's water resources, no matter their size (Liebe et al., 2005; Leemhuis et al., 2009). By increasing the availability of water during low-flow periods (International Commission On Large Dams, 2016), dams often play a key role in water supply, irrigated agriculture, hydropower generation, navigation, cattle breeding, fisheries, etc.

Despite these valuable applications, there is a scarcity of monitoring data as many countries cannot financially afford to build gauging stations (Solander et al., 2016). And even when monitoring systems do exist, there may not be institutions to collect the data, or legal means to disseminate it as it is often considered sensitive data (Alsdorf et al., 2007; Dombrowsky, 2007; Duan and Bastiaanssen, 2013). Yet this information is essential to conduct hydrological studies in committed basins, from defining



reservoir operation rules in simulation models (Yoon and Beighley, 2015), to assessing the impact of multi-reservoir systems on downstream river discharge (Vörösmarty et al., 1997; Hanasaki et al., 2006; Döll et al., 2009).

In that context, remote sensing is a promising tool to overcome the difficulty to access reliable information on a reservoir. This technique has also been applied to characterize a range of continental water bodies such as large lakes (Birkett, 1995; Ponchaut and Cazenave, 1998; Mercier et al., 2002), paddy rice fields (Islam et al., 2010) or tidal floods (Yan et al., 2010). The general procedure to monitor storage consists in associating water surface elevation and area after evaluating them independently (e.g., Frappart et al., 2006).

Satellite radar and laser altimetry are the predominant approaches to estimate the elevation of open water bodies (e.g., Morris and Gill, 1994; Crétaux and Birkett, 2006; Calmant et al., 2008; Gao et al., 2012; Wang et al., 2013), or their bathymetry (Arsen et al., 2014). Orbit repeat periods of radar altimeters such as Topex/Poseidon (T/P), GFO, Jason-1 and 2 or Envisat, range from 10 to 35 days. They have a high vertical accuracy with root mean square errors on the order of centimetres to tens of centimetres depending on the altimeter and the size of the water body (Calmant et al., 2008; Zhang et al., 2014). Yet, these sensors are affected by important drawbacks, including nadir viewing, narrow swath, coarse cross-track spacing (a few hundred kilometers), long along-track path length (about 1 km), large elevation differences around some water areas, that impede their application to more than a few hundred large lakes and reservoirs on the planet (i.e. area > 100 km$^2$ and width > 500 m) (Crétaux and Birkett, 2006; Alsdorf et al., 2007; Gao et al., 2012). Alternatively, the Geoscience Laser Altimeter System onboard the Ice, Cloud, and Elevation Satellite (ICESat/GLAS) measured land surface elevations between 2003 and 2009 with a much finer spatial resolution (footprints' size between 50 and 105 m every 170 m along track), a vertical accuracy close to 10 cm (Zhang et al., 2011; Duan and Bastiaanssen, 2013), and a finer cross-track resolution (15 km maximum at equator (Zwally et al., 2002)). There was however no continuous elevation retrieving: ICESat/GLAS gathered data only during designated campaigns, with a long ground-track repeat cycle for almost all of it (183 days). Furthermore, unlike radar altimeters that can be used under all weather conditions (Birkett and Beckley, 2010), laser measurements are affected by the presence of thin clouds (Duan and Bastiaanssen, 2013). Many existing studies consequently used ICESat/GLAS data to get a trend on pre-determined large lakes variations over several years (e.g., Zhang et al., 2011; Duan and Bastiaanssen, 2013; Song et al., 2013), or to calibrate area-elevation relationships for a limited number of water bodies large enough for the satellite to take sufficient elevation measurements per track (Zhang et al., 2014).

Water surface areas are commonly determined from optical satellite imagery such as MODerate Resolution Imaging Spectroradiometer (MODIS) and Landsat products (Xiao et al., 2006; Gao et al., 2012), or Synthetic Aperture Radar (SAR) sensors (e.g., RADARSAT, JERS-1 or ERS) (Annor et al., 2009; Duan and Bastiaanssen, 2013). The latter has however been less used due to the difficulty to get consistent results, as the required condition of a significantly lower phase coherence of water areas than of the surrounding land surface is not always met with orbital repeat cycles of more than a few days, or with wind or rain (Alsdorf et al., 2007; Eilander et al., 2014). Therefore, existing approaches have used either MODIS or Landsat depending on their emphasis on spatial or temporal resolution (Solander et al., 2016; Zhang et al., 2016). Images acquired during the various Landsat missions have a much finer spatial resolution (30 m) than MODIS's (250 m for the red band, 500 m for infrared), but they are taken on a repeat cycle of 16 days compared to the daily MODIS products. The higher revisit frequency of MODIS



satellites allows MODIS-based approaches to better address clouds and smoke artifacts on optical images. However MODIS missions cover a much shorter period (July 2000 to present) than Landsat missions (July 1982 to present).

The common protocol to separate water areas from other land use categories is to apply a threshold to indices such as the Normalized Difference Vegetation Index (NDVI) (e.g., Frappart et al., 2006; Gao et al., 2012), or the Modified Normalized Difference Water Index (MNDWI) proposed by Xu (2006) (e.g., Crétaux et al., 2015; Müller et al., 2016). But determining an adequate value for a multi-temporal analysis can be challenging because such a threshold is known to be case-dependent (Liu et al., 2012). Furthermore, separating water from land or vegetation may be difficult due to subpixel land-cover components (Ji et al., 2009), or water quality that can vary throughout a water body (Gao et al., 2012). To address these issues, decision tree defined thresholds have successfully been applied with various vegetation indices (e.g., Xiao et al., 2006; Islam et al., 2010; Yan et al., 2010), but remain case-dependent. Other methods like unsupervised classification (Wang et al., 2008), or direct elevation-area relationship from Digital Elevation Model (DEM, Wang et al., 2005) have also been tested but did not prove to be more precise. Gao et al. (2012) recently developed a method to combine both an index analysis and an unsupervised classification to improve the accuracy of the delineation of water areas. The approach was refined by Zhang et al. (2014) who enhanced the storage assessment with a novel surface area retrieval algorithm.

While promising, these approaches generally fail to systematically combine remote sensing surface area and elevation due to the different timing in orbital repeat cycles of different satellites. Elevation-area relationships are then deduced from remote sensing data that is available at the same time (e.g., through linear or polynomial regressions, Gao et al., 2012; Duan and Bastiaanssen, 2013; Song et al., 2013), so that reservoir storage can be computed with either remote sensing elevation or area only. Even then, existing methods estimate storage in relative terms, either from the already known elevation, area and storage at capacity (Zhang et al., 2014), or from the lowest water level detected (Duan and Bastiaanssen, 2013).

Furthermore, these approaches have only been applied to reservoirs larger than 100 km$^2$, which are estimated to represent only 0.54 % of reservoirs larger than 0.1 km$^2$ in the world (Lehner et al., 2011). Studies that analyzed small reservoirs delineated water surface with Landsat optical sensors (e.g., Liebe et al., 2005; Sawunyama et al., 2006; Rodrigues et al., 2012) or radar images to address the cloud cover issue (Annor et al., 2009; Liebe et al., 2009), and could only get an estimation of storage capacities by conducting bathymetrical surveys. Due to their reliance on *in situ* observations, these methods are inapplicable to remote, ungauged or conflict-prone areas.

This paper introduces a new method to monitor reservoir storage based on remote sensing data exclusively. The method is applied to small reservoirs (capacities and water surface areas starting from 1 hm$^3$ (million cubic metre) and 0.5 km$^2$ respectively) in the Yarmouk River Basin (YRB, see Fig. 1) in Southern Syria during the ongoing civil war and the decade before it started. Its prediction performance is tested against available *in situ* observations of reservoir storage and elevation in neighboring Jordan.

The document is organized as follows: Sect. 2 presents the method and algorithms developed for the monitoring of reservoir storage, Sect. 3 reviews results, error measurements and sensitivity analysis, and Sect. 4 concludes the study.





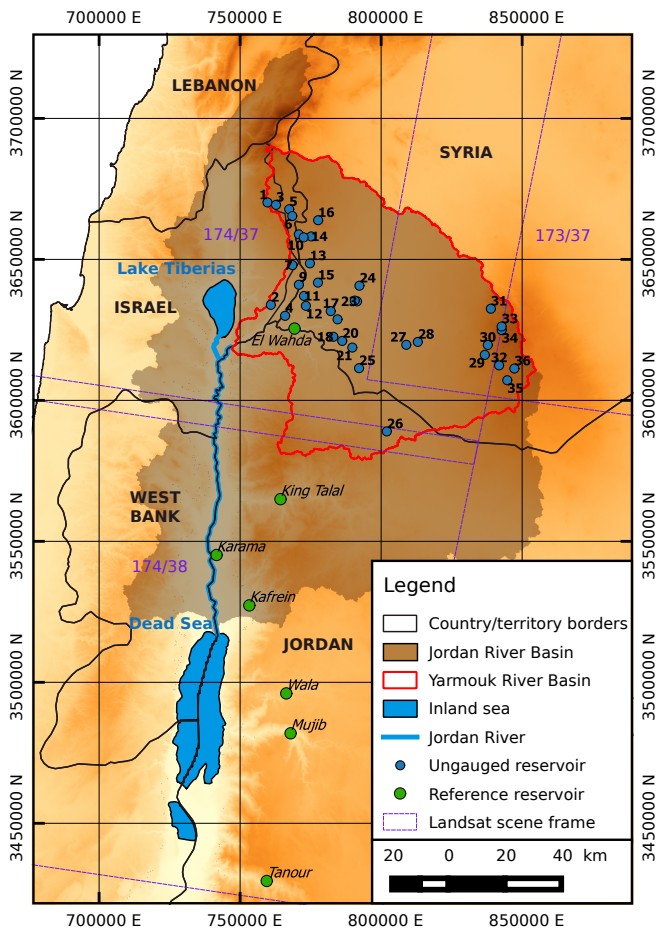

**Figure 1.** Reservoirs identified in Jordan and the Yarmouk River Basin using the method developed in this paper. Because *in situ* measurements are accessible for those managed by Jordan, they are used to validate the method. All coordinates are expressed in the Coordinate Reference System (CRS) WGS 84 / UTM zone 36N (EPSG:32636).

## 2 Methodology

The procedure is based on two types of data: Landsat images for water areas estimation, and DEM for topography. It works in three stages that are presented in the flowchart on Fig. 2. The idea behind the process is (i) to use Landsat bands to enhance the detection of water pixels, then (ii) to exploit this information to statistically correct the DEM vertical errors and characterize reservoir bathymetry, and (iii) to use the updated topography to reconstruct missing parts of Landsat images (e.g., pixels covered by clouds or not captured by the Landsat sensor).



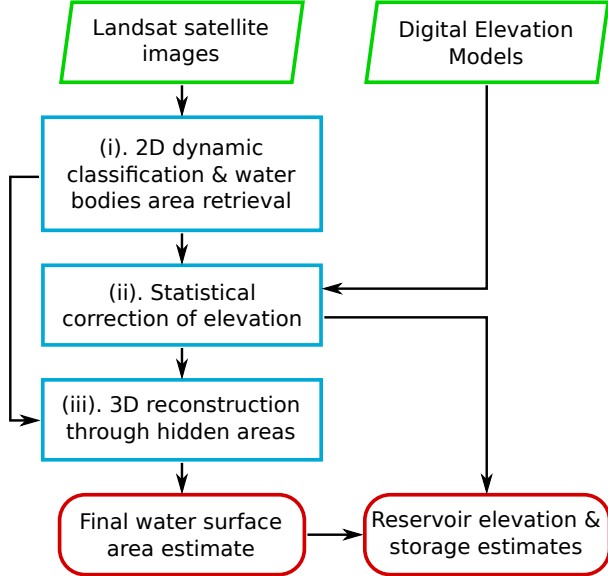

**Figure 2.** Flowchart of the whole procedure.

## 2.1 2D dynamic classification and water bodies area retrieval

Landsat images are chosen because they are freely available with a spatial resolution fine enough (30 m) to detect variations in the area of small reservoirs. The spatial resolution of MODIS images is indeed too coarse to assign to any small reservoir a proper range of area and elevation (1 km$^2$ is covered by 16 MODIS image 250 m pixels only). Thus, about 300 Landsat 4, 5,

7 and 8 images for each scene – index 173/37 above a part of the YRB, 174/38 above reservoirs in Jordan, and 174/37 above parts of both in the Worldwide Reference System (WRS, see the scene frames in Fig. 1) – are downloaded from the United States Geological Survey (USGS) EarthExplorer website (https://earthexplorer.usgs.gov/).

### 2.1.1 Fmask

We use the Fmask (Function of mask) algorithm (Zhu and Woodcock, 2012; Zhu et al., 2015) to discriminate cloud coverage

from open water. The algorithm was originally designed to separate potential cloud pixels from clear sky pixels on Landsat images using empirical thresholds on NDVI and the near-infrared band. Fmask distinguishes land and water areas and produce a probability mask for clouds, which we use to manually remove images that are almost entirely covered by clouds or with obvious large errors in water bodies detection. After quality control, about 245 images remain per location.

Most pixels classified as water by Fmask can reasonably be considered as water due to the relatively selective thresholds

used in the algorithm. Hence, at this stage, the uncertainty remains with regard to pixels hidden by clouds or cloud shadows, misclassified by Fmask as land or snow, or not captured by the Landsat sensor (e.g., "N/A" stripes caused by the Landsat 7



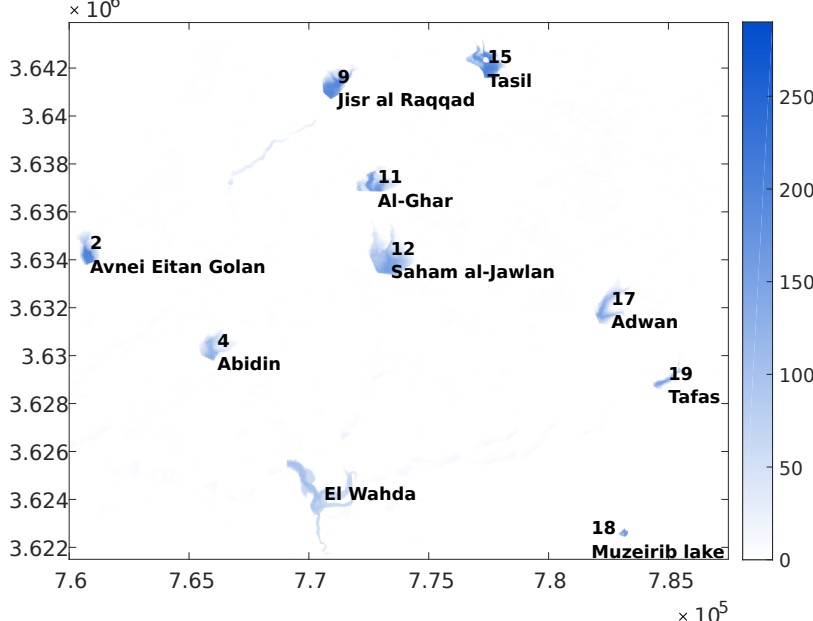

**Figure 3.** Image of the number of times each pixel has been covered by water ($M_{occ}$). Text in black indicates the identification number (for Syrian reservoirs) and name of known reservoirs. Coordinates are expressed in CRS WGS 84 / UTM zone 36N.

Scan Line Corrector (SLC) failure, see Fig. 4 (a) and (b)). Our analysis reveals that, on average, 24.1 % of reservoirs' pixels are misclassified as land, 8.1 % are covered with clouds or cloud shadows, and 8.6 % are in "N/A" areas (see Sect. 3.2).

### 2.1.2 Occurrence mask

We use the frequency with which pixels are classified as water to distinguish actual reservoirs from small pools or misclassified
5  land, and to delimit them. For each Landsat scene, the ~245 satellite images are superimposed to form an image where each pixel represents the number of times it has been covered by water (see Fig. 3). This occurrence mask ($M_{occ}$) is useful to filter occasional Fmask's classification errors, and to create a water mask ($M_{wat}$): pixels with values greater than 5 in $M_{occ}$ are classified as water and kept in $M_{wat}$, while those with lower values are considered as misclassified land and removed from water bodies (i.e. hidden by $M_{wat}$). In practice, the threshold of 5 was empirically chosen after comparing detected water bodies with
10  Google Earth high resolution (~1 m) imagery. The same threshold was applied to reservoirs located in the overlapping area of two Landsat images as it did not change their contours. Its small value is justified by the fact that most images with obvious mistakes have already been manually discarded at the previous step.

Aside from sporadic large wadis (intermittent rivers) that are manually removed from the mask, final water bodies in $M_{wat}$ are deemed to be reservoirs. They are the ones depicted in blue and green dots on the map in Fig. 1.





### 2.1.3 Classification enhancement for each Landsat image

The detection of water bodies is enhanced using NDVI and MNDWI rasters computed from Landsat imagery. A low NDVI can be attributed to both water and bare land, and a low MNDWI value can denote either water or clouds. We combine these indices and leverage their complementary nature to detect open water.

To ensure more reliable and repeatable values for identical land use categories in different images, the two indices are computed from Top of Atmosphere (TOA) reflectance to which the imaged-based atmospheric correction Dark Object Subtraction 1 (DOS1, Chavez, 1996) has been applied to estimate surface reflectance. However the DOS1 adjustment is not optimal because not based on actual atmospheric or cloud cover measurements. Moreover, the slight band variations between the various Landsat missions may affect NDVI and MNDWI, and may require different thresholds to detect water. Consequently, two

supplementary water detection adjustments are performed through the method presented in the flowchart on Fig. 5 to define a MNDWI threshold adapted to each date and climatic conditions (i.e. each time $t$ over a given scene). A NDVI mask ($\mathbf{M_{NDVI}}(t)$) is first created to calibrate the MNDWI threshold, which is then used to build a MNDWI mask representing water areas ($\mathbf{M_{MNDWI}}(t)$):

1. The goal of $\mathbf{M_{NDVI}}(t)$ is to find with NDVI all pixels where there could potentially be water. Depending on the results of

the Fmask classification in $\mathbf{M_{wat}}$, three situations can arise:

(i) If water is already detected by Fmask in $\mathbf{M_{wat}}$ reservoirs, $\mathbf{M_{NDVI}}(t)$ is formed from those ones (see green dots in the Fig. 4 (c) example).

(ii) Where water is not detected by Fmask, we impose a threshold on the NDVI. Pixels with a NDVI lesser than –0.1 in $\mathbf{M_{wat}}$ are used to form $\mathbf{M_{NDVI}}(t)$. The lowest values are indeed generally typical of water. This condition has been added

to take into account images where thin clouds cover reservoirs.

(iii) In ∼1 % of all images, no pixel meets the previous conditions, even if water can be seen by eye on the original spectral bands. We have indeed noticed that for these few cases, Fmask associates water bodies to clouds. $\mathbf{M_{NDVI}}(t)$ is thus built on pixels classified as cloud in $\mathbf{M_{wat}}$.

2. The detection of water bodies is further enhanced by imposing a threshold on MNDWI images. The threshold is defined

automatically so as to optimally distinguish water bodies from clouds. For the two first situations, the threshold is set to the $X_{MNDWI}$th percentile ($P_{X_{MNDWI}}$) of MNDWI values in $\mathbf{M_{NDVI}}(t)$. $X_{MNDWI}$ is set to the maximum, 100, in order to avoid over-constraining the classification. The sensitivity of the method to the choice of this parameter is presented in Sect. 3.2 further below. For the third case, the threshold is set to the $70^{th}$ percentile ($P_{70}$) of MNDWI values in $\mathbf{M_{NDVI}}(t)$.

Finally, the water mask – or MNDWI mask – $\mathbf{M_{MNDWI}}(t)$ is formed by including only areas with a MNDWI lesser than

the MNDWI threshold in $\mathbf{M_{wat}}$. This last step allows to incorporate most water pixels left out by Fmask and undetectable with NDVI (see red dots in Fig. 4 (d)).





**Figure 4.** 2D dynamic water classification over a part of a Landsat 7 image (174/37) obtained on March 30[th], 2010. (a) SWIR-R-G image. Two reservoirs can be seen by eye – even if their appearance is very similar to cloud shadow areas –, but the hedges are not easy to detect due to the cloud cover. (b) Results of the Fmask classification. Water areas detection is not precise enough to directly use the results for the estimation of reservoirs surface area. (c) NDVI image. Water pixels' low NDVI here contrasts with the surrounding irrigated crops' high NDVI, as the two reservoirs are located close to cultivation areas. (d) MNDWI image. Red dots indicate water areas obtained after the 2D enhancement (Sect. 2.1). The 3D reconstruction is operated later (Sect. 2.3) on the *Unknown* part.





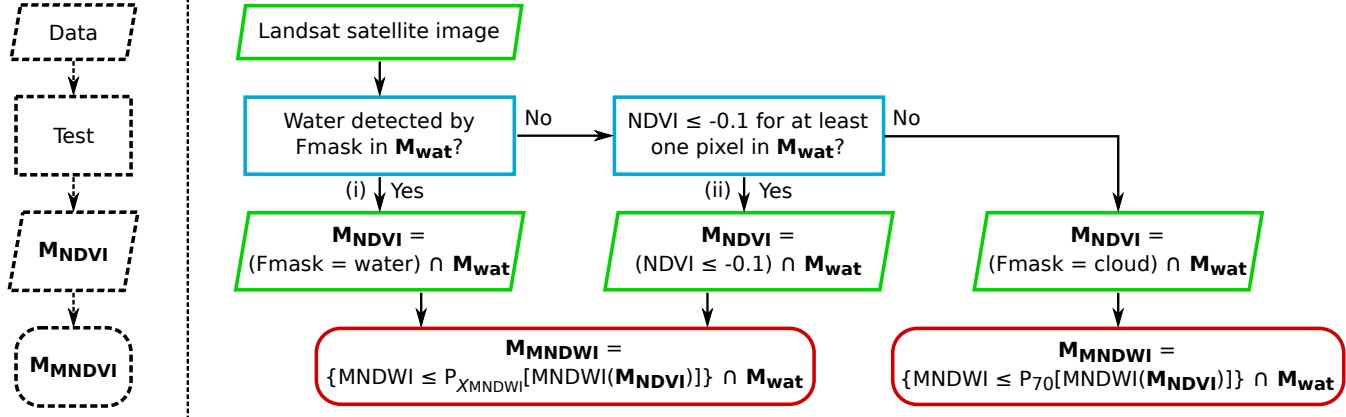

**Figure 5.** 2D dynamic classification procedure.

After removing water areas smaller than 20 pixels ($20 \times 900$ m$^2$), considered noise, the classified images have three categories: (i) *Water* as identified by the protocol developed above, (ii) *Land* according to Fmask, and if not in the category *Water*, and (iii) *Unknown* that include all other pixels (see Fig. 4 (d)).

## 2.2 Statistical correction of elevation

### 2.2.1 Digital elevation models

Unlike most studies, the proposed method does not rely on satellite altimetry to assess water bodies' elevation, but on DEMs to get the topography. It is then required that reservoirs were almost empty or not yet built when the DEM satellites passed over them, for at least one of the two sources considered: ASTER GDEM v2 and SRTM-C/X. ASTER GDEM v2 data was acquired between 2001 and 2008, and SRTM-C/X data on February 11–22, 2000. All have a spatial resolution of 1" (approximately 30 m at the equator) which we resample to match Landsat images. The large coverage of these datasets is chosen over the very low precision of the measures. They indeed cover almost all Earth's land surface (except for SRTM-X): from 83° N to 83° S for ASTER GDEM v2, and from 60° N to 56° S for SRTM-C; but the vertical relative precision is very low compared to satellite altimetry: objectives of 15 m and 6 m for 90 % of SRTM-C and SRTM-X data respectively (German Aerospace Center (DLR), 2017; Rodriguez et al., 2005), and standard deviations estimated to 3.95 m and 8.68 m for SRTM-C and ASTER data respectively (ASTER GDEM Validation Team, 2011).

### 2.2.2 Elevation-area relationship

To improve elevation assessment, DEMs are statistically corrected by using the information on water surface areas obtained from Landsat images. The protocol presented in Fig. 6 is implemented for each reservoir:



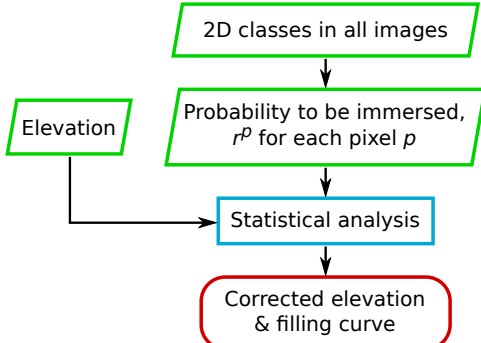

**Figure 6.** Procedure for the statistical correction of topography.

1. A water coverage quantile is computed at each pixel to determine the probability for it to be immersed. To each pixel $p$ is associated the ratio $r^p$ defined as:

$$r^p = \frac{N^p_{water}}{N^p_{water} + N^p_{land}} \tag{1}$$

   where $N^p_{water}$ is the number of times the given pixel $p$ is counted as Water, and $N^p_{land}$ the number of times it is counted as Land. We ignore the images where the pixel $p$ is classified as Unknown.

2. Each pixel's elevation $H^p$ is put in relation with the area $A^p$, defined as the cumulated area of all pixels $q$ for which $r^q \geq r^p$. The examples of Fig. 7 suggest that pixels' elevation is not always correlated with the number of times they are classified as water. To a certain extent this was expected from the DEM's low vertical precision, but some "anomalies" concerning the most often immersed pixels (i.e. lowest $A^p$) can be recurrent from one reservoir to another due to either a strong dispersion in elevation (see SRTM-X data in Fig. 7 (b)), or a flat elevation (see SRTM-C data in Fig. 7 (c)). We interpret this irregularity as arising from the presence of water where the satellite tried to evaluate elevation: in the case of SRTM-X, the measure over water is hampered for reasons inherent to the use of a SAR sensor; and in the case of SRTM-C, DEM pixels covered with water may have been filled during a post-treatment analysis. Either way, elevation cannot be retrieved from the given DEM for these reservoirs' most often immersed pixels.

3. To address the issue, a polynomial regression on observed land pixels ($A > A_i$, with $A_i$ the area assumed as immersed during the satellite elevation retrieval) is used to build a "corrected elevation"-area relationship ($A \rightarrow H_c(A)$) that smoothes the data and filters potential errors. Extreme values of $H$ are ignored. This step is executed three times – one for each DEM – and the better quality dataset is kept. A few examples are shown in Fig. 7.

4. A filling curve – volume-area relationship – is finally constructed using the outcomes of the previous step.

The regression relies on the assumption that elevation estimates are correct on average by considering many pixels. Indeed, the relative error on elevation approaches zero when the number of images taken into account grows. This property has already been used by LeFavour and Alsdorf (2005) for instance, in order to estimate the slope of the Amazon River.





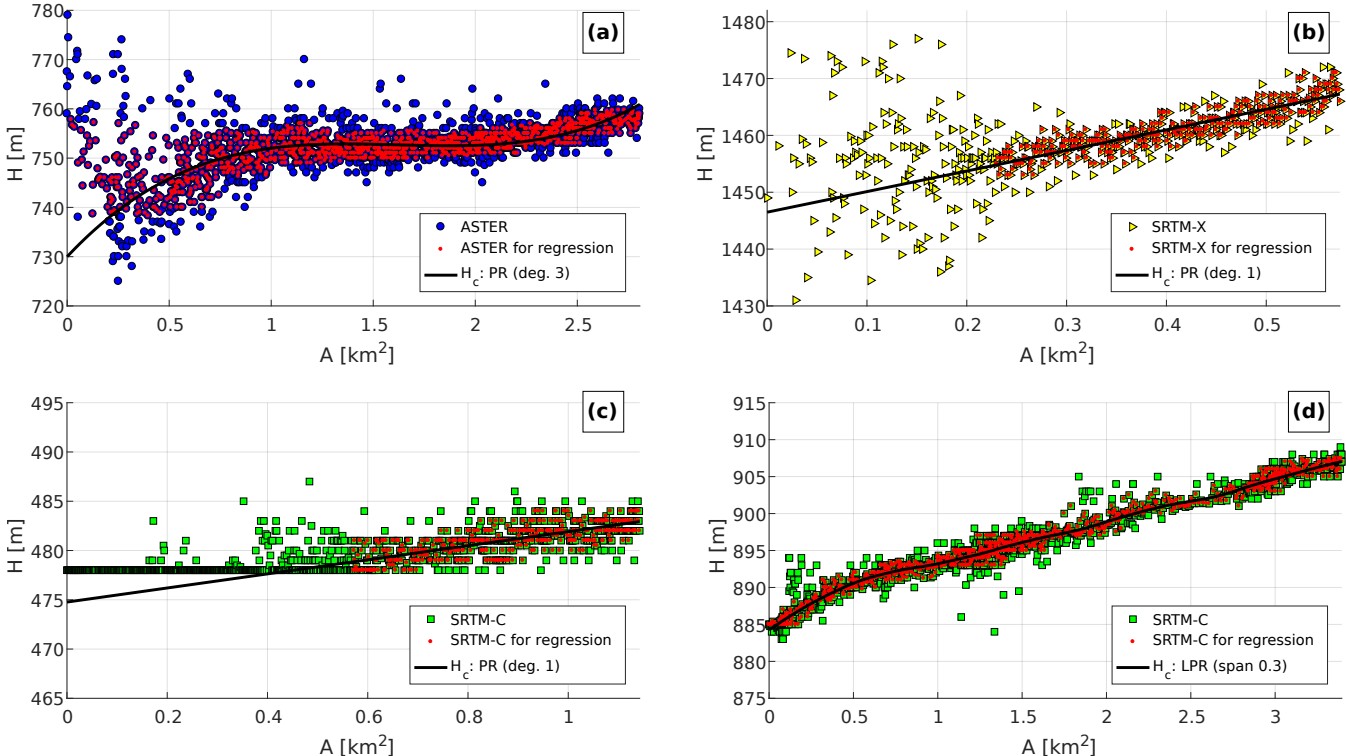

**Figure 7.** Elevation-area relationship and regression for a few reservoirs in the Yarmouk River Basin: (a) Kudnah, (b) Roum, (c) Al Ghar and (d) Qunaitera. Each symbol (circle, square or triangle depending on the DEM) represents the information associated to one pixel in a reservoir. $PR$ and $LPR$ stand for Polynomial Regression and Local Polynomial Regression respectively.

Parameters and results of the regression for reservoirs that fulfill the criteria mentioned at the beginning of this article – maximum storage and area larger than 1 hm$^3$ and 0.5 km$^2$ respectively – are summarized in Table 1.

## 2.3  3D reconstruction through hidden areas

Retrieving missing parts of water bodies in the Unknown areas means dealing with Landsat drawbacks: (i) the 16 days repeat
5  cycle making images regularly covered by clouds, and (ii) the failure of the Landsat 7 SLC that led to large data losses for the
Enhanced Thematic Mapper Plus (ETM+) sensor after May 2003 (see gray stripes in Fig. 4 (b)).

Zhang et al. (2014) developed an approach to improve quite significantly the estimation of reservoir's water area. However, their method requires that only a small part of the reservoir is misclassified or hidden. This is not a problem if one works with MODIS images over very large reservoirs, but in our situation – Landsat images over small water bodies – the condition is
10  rarely met.

We developed an alternate algorithm to use the information from each individual pixel:





| Location | Reservoir | DEM | Visible area $1 - \frac{A_i}{A_{max}}$ [%] | Regression | $R^2$ | $A_{max}$ [km$^2$] | $\Delta H_{c\,max}$ [m] | $V_{max}$ [hm$^3$] |
|---|---|---|---|---|---|---|---|---|
| Israel-controlled | Al Manzarah | ASTER | 100 | PR (deg. 2) | 0.34 | 0.53 | 9.14 | 2.64 |
| Golan Heights | Avnei Eitan Golan | ASTER | 70 | PR (deg. 2) | 0.31 | 0.93 | 4.88 | 2.34 |
| | Abidin | ASTER | 65 | PR (deg. 1) | 0.37 | 1.16 | 8.74 | 5.07 |
| | Qunaitera | SRTM-C | 100 | LPR (span 0.3) | 0.98 | 3.40 | 22.81 | 33.94 |
| | Jisr al Raqqad | ASTER | 30 | PR (deg. 1) | 0.52 | 1.16 | 16.23 | 9.43 |
| | Kudnah | ASTER | 100 | PR (deg. 3) | 0.46 | 2.81 | 30.92 | 29.45 |
| | Al Ghar | SRTM-C | 50 | PR (deg. 1) | 0.56 | 1.14 | 8.17 | 4.66 |
| | Saham al-Jawlan | SRTM-C | 55 | PR (deg. 1) | 0.84 | 2.48 | 12.93 | 15.99 |
| Syria | Ghadir al-Bustan | ASTER | 50 | PR (deg. 1) | 0.56 | 1.19 | 15.02 | 8.93 |
| | Tasil | ASTER | 60 | PR (deg. 1) | 0.28 | 1.28 | 9.59 | 6.15 |
| | Adwan | ASTER | 100 | PR (deg. 1) | 0.33 | 1.31 | 7.92 | 5.17 |
| | Ebtaa kabeer | SRTM-C | 80 | PR (deg. 1) | 0.71 | 0.73 | 6.56 | 2.39 |
| | Sheikh Miskin | SRTM-C | 45 | PR (deg. 1) | 0.71 | 2.85 | 7.51 | 10.71 |
| | Roum | SRTM-X | 60 | PR (deg. 1) | 0.81 | 0.57 | 20.77 | 5.94 |
| | Sahwat al-Khadr | SRTM-C | 80 | PR (deg. 3) | 0.78 | 1.27 | 10.07 | 6.49 |
| Border Jordan-Syria | El Wahda | SRTM-C | 100 | LPR (span 0.3) | 0.97 | 2.69 | 53.31 | 66.72 |
| | Karama | SRTM-C | 85 | LPR (span 0.1) | 0.90 | 3.79 | 17.00 | 35.91 |
| | Kafrein | SRTM-C | 30 | PR (deg. 1) | 0.56 | 0.66 | 17.80 | 5.85 |
| | Tanour | SRTM-C | 85 | PR (deg. 1) | 0.94 | 0.59 | 36.00 | 10.56 |
| Jordan | King Talal | SRTM-C | 20 | PR (deg. 1) | 0.29 | 2.17 | 31.66 | 33.69 |
| | Wala | SRTM-C | 100 | LPR (span 0.5) | 0.85 | 0.61 | 25.86 | 6.37 |
| | Mujib | SRTM-C | 50 | LPR (span 0.3) | 0.79 | 1.30 | 44.33 | 30.49 |

**Table 1.** Parameters and results of the elevation-area regression. $PR$ and $LPR$ stand for Polynomial Regression and Local Polynomial Regression respectively. $R^2$ is the coefficient of determination between the corrected elevation $H_c$ and the elevation $H$ for pixels taken into account by the regression (red dots in Fig. 7).

1. As the area $A^p$ has been associated to each pixel $p$, and $H_c$ has been expressed in terms of $A$, a corrected elevation is associated to each pixel in a reservoir.

2. Each pixel in an Unknown area adjacent to water areas is set to Water if: (i) the pixel is in $\mathbf{M_{wat}}$, and (ii) its corrected elevation $H_c^p$ is lesser than the $X_{H_c}^{\text{th}}$ percentile of corrected elevation in all adjacent water bodies. This threshold is set to 98 to ignore highest values of $H_c$, in case they were associated to pixels misclassified as water. A sensitivity analysis has been conducted with regard to this threshold, and the results are available in Sect. 3.2 further below.





This water body reconstruction technique relies on the fact that a pixel that is often immersed likely has an elevation lower than a pixel that is rarely immersed. This is a reasonable assumption due to the large number of images analyzed. Blue dots in Fig. 8 show how the 3D reconstruction complements the previous 2D information retrieval. Finally, storage variations are obtained by combining final reconstructed areas with the previously determined filling curves.

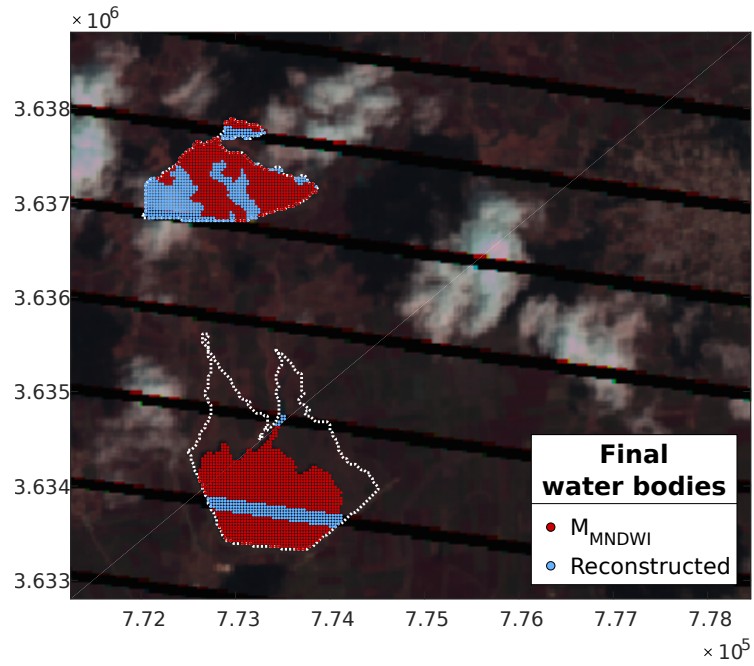

**Figure 8.** SWIR-R-G image. Final water bodies as obtained after the 2D enhancement and the 3D reconstruction applied to the Landsat 7 image (174/37) taken on March 30[th], 2010 (same as Fig. 4).

## 3  Results

### 3.1  Storage variations: validation and discussion

Storage variations estimated by remote sensing for all reservoirs that cannot be gauged in the YRB are displayed in Fig. 9. These reservoirs are located in Syria and in the Israel-controlled Golan Heights. We can see coherent storage variations through the presence of drawdown-refill cycles, which means that the 2D enhancement and 3D reconstruction steps have improved the detection of water and helped to overcome the low Landsat repeat cycle of 16 days.

Reservoirs managed by Jordan are used to validate the method by comparing our remote sensing estimates of elevation and storage with *in situ* measurements conducted by the Jordan Valley Authority (JVA). With the exception of the King Talal dam, our results seem to follow quite accurately the historical records (see Fig. 10). In addition, we can note that elevation $H$ and volume $V$ may vary a lot from month to month: up to 10 m or 15 hm$^3$ – i.e. 50 % of the maximal storage – for instance for





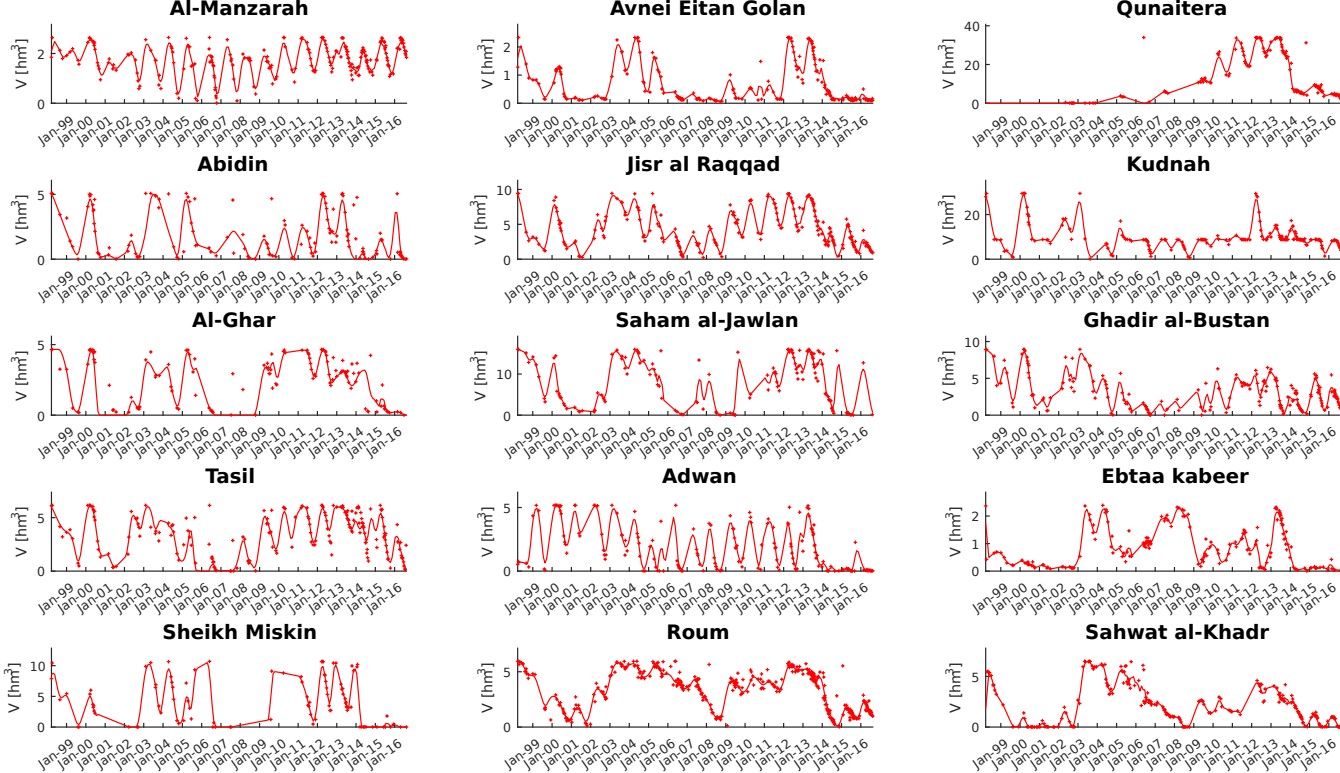

**Figure 9.** Storage variations. Red crosses indicate estimates from the method developed in this paper. Red lines are local polynomial regressions, that are plotted only with the purpose of showing storage variation trends.

the Mujib reservoir. Some of the differences between our estimates and measured data might then come from the inaccuracy regarding the data collection date.

With regard to the King Talal reservoir, we can see large errors in storage estimates (see Fig. 10). But they could have been expected at the end of the elevation-area relationship establishment step: the assumptions that were made to define $H_c$
5   were maybe not justified in this case. Indeed, 80 % of the reservoir maximal area was covered with water when the SRTM satellite passed over the dam, and the $R^2$ is only 0.29 for the regression applied to the remaining visible pixels (see Table 1). A small visible surface area does not necessarily lead to a low quality elevation-area relationship – see the good estimates for the Kafrein reservoir, while 70 % of its maximal area was hidden when the SRTM satellite passed over it –, but it certainly is a sign that results might be biased.

10   Errors on the estimation of elevation and storage are evaluated in terms of the coefficient of determination ($R^2$, Eq. 2) and the average relative error ($\varepsilon_m$, Eq. 3):

$$R^2 = \frac{Cov(RS, Hist)^2}{\sigma_{RS}^2 \cdot \sigma_{Hist}^2} \tag{2}$$





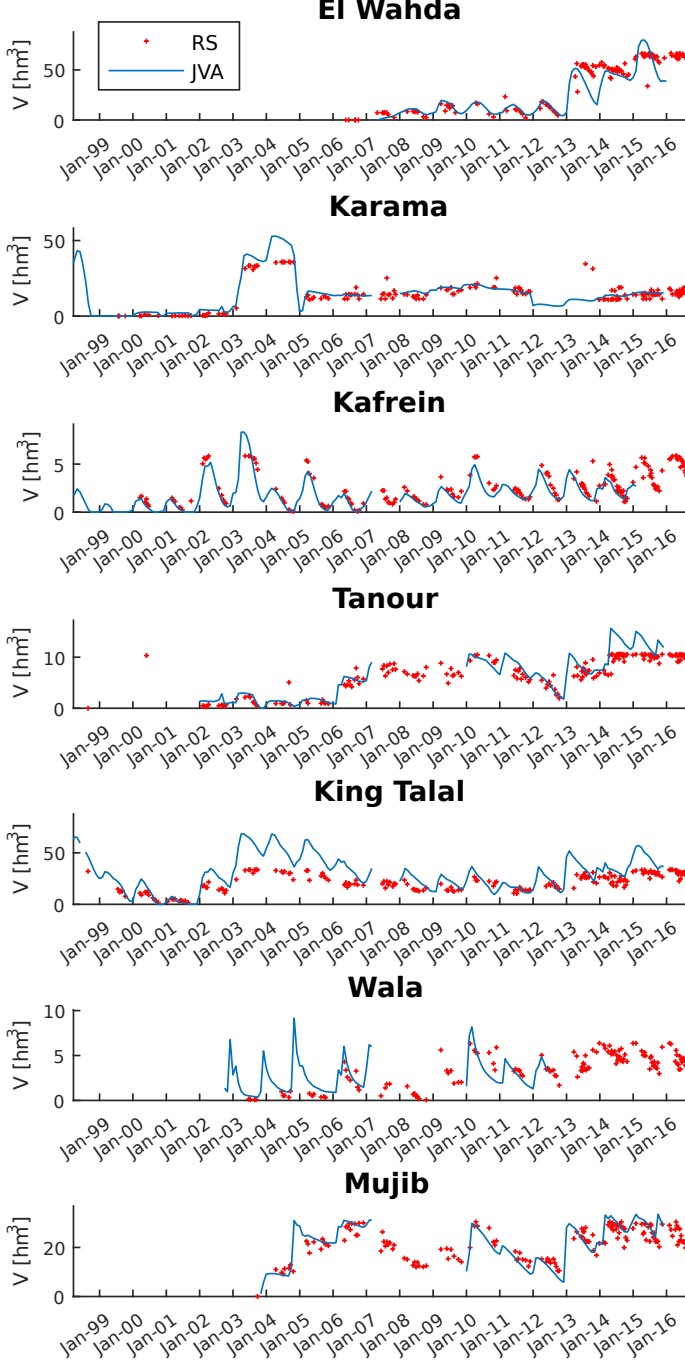

**Figure 10.** Storage variations for Jordan managed reservoirs. Red crosses indicate estimates from the method developed in this paper. The blue lines indicate *in situ* data records that were made by the Jordan Valley Authority (JVA).





$$\varepsilon_m = \frac{1}{N} \sum_{i=1}^{N} \frac{(RS_i - Hist_i)}{Hist_i} \tag{3}$$

where $Cov(RS, Hist)$ is the covariance between remote sensing (RS) estimates and JVA historical measurements, $\sigma^2$ the variance, and $N$ the number of RS estimates during the period that JVA measured storage or elevation. Results are presented in Table 2.

| Reservoir | N | | $R^2$ | | $\varepsilon_m$ | |
|---|---|---|---|---|---|---|
| | $H_c$ | $V$ | $H_c$ | $V$ | $H_c$ | $V$ |
| El Wahda | 25 | 107 | 0.54 | 0.76 | 0.49 | 0.20 |
| Karama | 29 | 123 | 0.98 | 0.79 | −0.00 | −0.13 |
| Kafrein | 35 | 136 | 0.91 | 0.81 | 0.00 | 0.35 |
| Tanour | 16 | 117 | 0.83 | 0.84 | 0.09 | −0.12 |
| King Talal | 40 | 159 | 0.50 | 0.76 | 0.87 | −0.15 |
| Wala | 15 | 37 | 0.36 | 0.69 | 0.27 | −0.13 |
| Mujib | 15 | 104 | 0.73 | 0.75 | −0.18 | 0.03 |

**Table 2.** Errors in terms of $R^2$ and average relative error for Jordanian reservoirs' $H_c$ and $V$ assessments.

The coefficient of determination for storage ranges from 0.69 to 0.84. These high values confirm an important correlation and the similar variation trends that can be seen between the method's estimates and JVA records (see Fig. 10). A few high $|\varepsilon_m|$ values for both $V$ and $H_c$ though indicate that there is still some uncertainty with regard to the estimation of their absolute value at a given month. Indeed, by ignoring the King Talal dam, $|\varepsilon_m|$ ranges from 3 % to 35 % for storage, and reaches up

to 49 % for elevation. These error estimates for elevation though need to be taken into account with caution due to the small number of JVA measurements available for comparison (15 < N < 35).

In order to better evaluate the proposed method compared to a basic fixed NDVI and near-infrared thresholds water area detection, we consider the results presented in Table 3: on average, only 30.0 % to 59.4 % of final reservoir areas are detected by Fmask. The average additional part of final water bodies that is detected with the employment of a NDVI-based dynamic

threshold for MNDWI is larger than 30 % for all Jordan reservoirs, and can reach more than 50 % for Tanour and Wala reservoirs. Similarly, the average additional part obtained through the 3D reconstruction is larger than 3.9 % (Karama reservoir), and goes beyond 16 % for the more recent reservoirs Tanour, Wala and Mujib, whose construction ended after 2002 – proportionally, more Landsat 7 images affected by "N/A" stripes were then used for them than for older dams. In light of these large shares of hidden or undetected water areas, corrections were obviously essential to consistently monitor reservoirs elevation

and storage.





| Reservoir | Fmask classification [%] | | | | Changes [%] | |
|---|---|---|---|---|---|---|
| | Water | Land | Other | N/A | 2D | 3D |
| El Wahda | 58.6 | 20.8 | 13.1 | 7.5 | 32.2 | 9.2 |
| Karama | 64.1 | 13.3 | 20.9 | 1.7 | 32.0 | 3.9 |
| Kafrein | 58.5 | 15.9 | 17.2 | 8.4 | 31.9 | 9.7 |
| Tanour | 31.3 | 15.4 | 39.0 | 14.3 | 52.5 | 16.1 |
| King Talal | 59.4 | 22.1 | 9.7 | 8.8 | 30.8 | 9.8 |
| Wala | 30.0 | 24.4 | 30.0 | 15.7 | 52.6 | 17.5 |
| Mujib | 36.1 | 9.6 | 37.2 | 17.2 | 45.2 | 18.6 |

**Table 3.** Initial Fmask classification inside the final water areas ("Other" refer to clouds, cloud shadows and snow), and stages' percentage changes that led to the classification as water ("2D" for the 2D classification enhancement, and "3D" for the 3D reconstruction).

## 3.2 Sensitivity analysis

The two algorithms used to improve the estimation of reservoirs area rely on one empirical threshold each: the classification enhancement is performed through the definition of a MNDWI percentile threshold ($X_{\mathrm{MNDWI}}$) to build a mask dynamically adapted to each Landsat image, and the reconstruction is achieved with the choice of a percentile for $H_c$ values ($X_{H_c}$), which

is set to avoid water areas overestimation.

The sensitivity of the whole method to these two parameters is tested in terms of the above defined indices: $R^2$ and $\varepsilon_m$, that are averaged for all reservoirs in Jordan (King Talal excluded). The sensitivity analysis is conducted by making the percentile thresholds vary between 90 and 100 with a step of 1. Results are presented for both storage and elevation in Fig. 11.

The coefficient of determination reaches its maximum with $X_{\mathrm{MNDWI}}$ values around 98 for storage and 93 or 95 for elevation.

However, $R^2$ does not quantitatively assess the accuracy of the method, and as it remains fairly high (above 0.78 for storage, or 0.74 for elevation) in the whole 90–100 range for both parameters, it is not considered to select the threshold percentiles.

In addition, $|\varepsilon_m|$ decreases as $X_{\mathrm{MNDWI}}$ and $X_{H_c}$ increase. The method does not detect an excessive number of water pixels – see the retrieval over the large missing parts detailed in Table 3 –, but rather obtains estimates for elevation and storage closer to the measurements conducted by JVA. Two conclusions can be drawn from these observations. First, the success in

the 2D enhancement means that there is enough information in Landsat bands to better detect water areas. And second, the precision of the 3D reconstruction implies that enough Landsat images are available for most reservoirs to statistically improve the detection of water bodies when clouds or "N/A" stripes hide land.

However, the $H_c$ upper limit for the reconstruction has a decreasing impact on $\varepsilon_m$ as the MNDWI threshold increases: fewer missing water pixels leads to fewer pixels available to "fill with water" during the subsequent reconstruction. For lower

$X_{\mathrm{MNDWI}}$ values, the decrease in $|\varepsilon_m|$ for high $X_{H_c}$ values is clearer. It shows that the reconstruction algorithm addresses well the Fmask and dynamic threshold method limitations, even if it cannot entirely balance the errors. The fact that $|\varepsilon_m|$ is on average





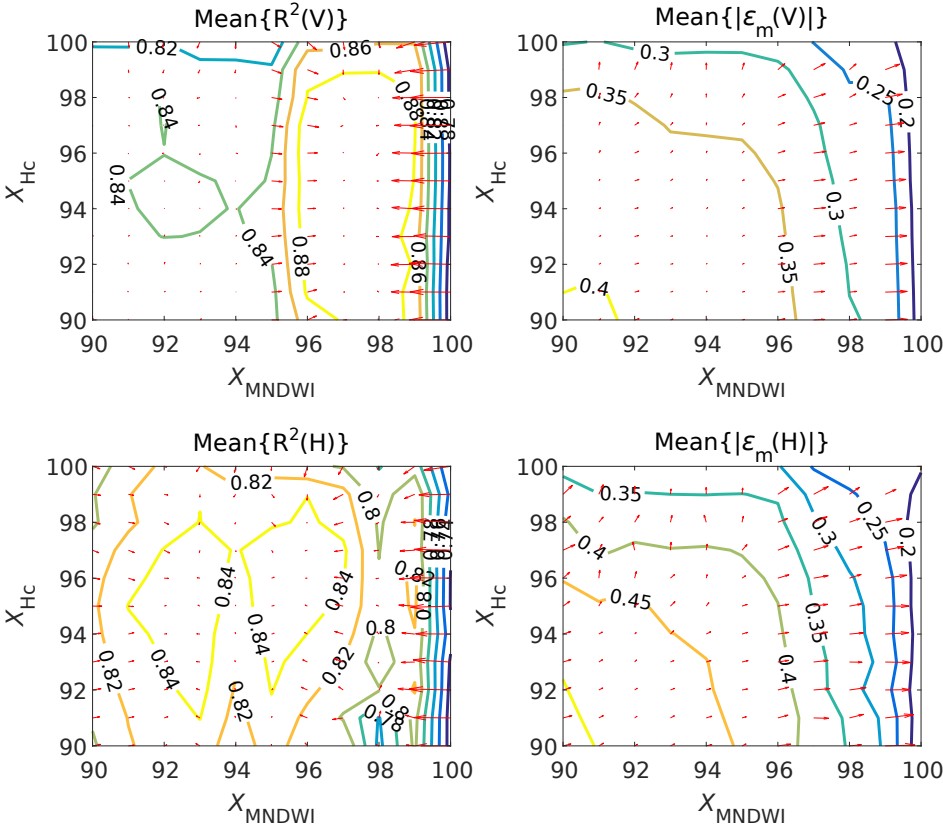

**Figure 11.** Sensitivity analysis of the two thresholds used to improve water bodies surface area estimates. Indices are averaged for all Jordanian reservoirs (except King Talal). Arrows point towards larger $R^2$, or lower $|\varepsilon_m|$, and lengths are proportional to the gradient.

lower for maximal $X_{H_c}$ values than for maximal $X_{\mathrm{MNDWI}}$ values could however be expected as the reconstruction relies on the reservoir's elevation-area relationship, which is established from the elevation of the pixels that are detected in the first stage.

In the end, the percentiles that we chose in this study – respectively 100 and 98 for $X_{\mathrm{MNDWI}}$ and $X_{H_c}$ – enable a trade-off between the options of lowering $|\varepsilon_m|$ for both storage and elevation. Also, with these percentiles, $R^2$ is still significantly high to ensure a strong correlation. It should be noted that the thresholds do not depend on the location, nor the date the Landsat images were taken. Therefore, the sensitivity analysis reveals that highest values for both $X_{\mathrm{MNDWI}}$ and $X_{H_c}$ could be used to apply the method to any other region in the world.

## 4 Conclusions

Although information on small reservoirs storage is crucial for water management in a river basin, it is most of the time not freely available in remote, ungauged or conflict-prone areas. A remote sensing method is proposed in this paper to monitor



small water bodies (capacities and water surface areas starting from 1 hm$^3$ and 0.5 km$^2$ respectively). The method is based only on DEMs for elevation, and Landsat satellite images for water surface area, to quantitatively estimate storage variations.

The method is applied to reservoirs in Syria and the Israel-controlled Golan Heights in the Yarmouk River Basin, and an uncertainty analysis is conducted with neighboring Jordan reservoirs for which *in situ* measurements are available. The average
relative error is relatively low compared to the size of the studied reservoirs and the precision of the datasets that are used.

The main limitation of the approach is its inapplicability to reservoirs that were significantly "covered" with water when the DEM satellites passed over them. Fortunately, this information can be readily obtained from remote sensing data and used to determine the applicability of the method *a priori*.

For all "uncovered" small or large reservoirs, the uses of datasets available over the whole continental surface, and of
thresholds dynamically defined for both the 2D enhancement and the 3D reconstruction, make the method potentially suitable to monitor reservoirs in inaccessible areas. Moreover, the precision of the filling curve and the 3D reconstruction algorithm increases with the number of pixels taken into account. Applying the method to large "uncovered" reservoirs could then potentially lead to better results. The sensitivity analysis also shows that choosing maximum thresholds in both water area retrieval stages gives the best reservoir storage estimates.
Furthermore, the algorithms used in the method automatically detect water bodies, define the water areas retrieval parameters, build filling curves and assess reservoir storage. They could thus provide near real-time updates on water bodies storage. Indeed, Landsat images are produced every 16 days (or 8 days for parts covered by several scenes with different WRS), and each new image provides additional information to the algorithmic tools that can then learn by themselves and correct previous estimates while generating new ones.

*Data availability.* Underlying research data are not publicly accessible. Remote sensing data access for this study is explained in Sect. 2. JVA data records are not publicly available.

*Author contributions.* N.A. and A.T. developed the method and conducted the case study; N.A., A.T., M.F.M. and H.Z. analyzed data; and N.A. wrote the paper with contributions from all co-authors.

*Competing interests.* The authors declare that they have no conflict of interest.

*Acknowledgements.* Jordanian reservoirs monitoring data were provided by Jordan's Ministry of Water and Irrigation, and Jordan Valley Authority. Landsat satellite images were obtained through the United States Geological Survey (USGS) EarthExplorer (https://earthexplorer. usgs.gov/). ASTER GDEM is a product of METI and NASA, and was found on EarthExplorer. SRTM (C-band) data were released by NASA, and are available at the US Geological Survey's EROS Data Center (https://eros.usgs.gov/). SRTM/X-SAR were operated by the





German Aerospace Center (DLR) with participation of the Italian Space Agency (ASI), and obtained through EOWEB (http://eoweb.dlr.de:8080/index.html). This work was conducted as part of the Belmont Forum water security theme for which coordination was supported by the National Science Foundation under grant GEO/OAD-1342869 to Stanford University. Any opinions, findings, and conclusions or recommendations expressed in this material are those of the authors and do not necessarily reflect the views of the National Science Foundation.

5   The authors acknowledge the financial support of NSERC through grant G8PJ-437384-2012.





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
