# Peer review of "Monitoring small reservoirs storage from satellite remote sensing in inaccessible areas"

_Hydrology and Earth System Sciences, 2017_

## Short Comment (SC1) · 31 Jul 2017

The manuscript by Avisse et al. describes an elaborated approach for retrieving the storage volumes of smaller reservoirs from remote sensing. As it relies exclusively on well-available optical and DEM data, it seems a valuable contribution for the monitoring of these storages in data-scarce regions. Since the authors also emphasise the general usefulness und transferability in this regard, I'd like to encourage them to share the required source code of the algorithm, which would match the spirit of publishing in an Open-Access-journal.

Further minor suggestions: - Fig. 10 suggests that the methods tends to underestimate large volumes. Especially for Karama and Tanour there seems to be a upper limit,

[Figure]

which the predictions of the method do not exceed. This is apparently not related to the complete filling of the reservoirs, as the ground observations confirm some dynamics within these phases. Is there any explanation to that?

- Table 2: The values of eps_m(V) for Kafrein and King Talal differ surprisingly from the impression one gets in Fig 10: In the plot, Kafrein seems to be modelled much better than King Talal. Is there any explanation for this surprising impression?

- Specifying a relative error for H (eps_m(H), Table 2, Fig.11) does not make sense to me: If H is water surface elevation, eps_m will then depend on absolute altitude. Instead, water level (H - H_min) or absolute deviation (H_RS-H_HIST - mean(H_RS-H_HIST)) should be used.

- The choice of the regression used for reconstructing the H-A-relationship is not explained: According to Tab. 1, "Polynomial Regression" of different order and "Local Polynomial Regression" are used. Are they selected by best fit? The respective description (p. 10, ll. 15) is quite vague, especially concerning the 3-fold repetition of the process and the exclusion of outliers.

- When discerning water surfaces, water bodies with macrophyte growth remain a serious challenge. It would be interesting to discuss if the presented approach for eliminating the SLC-data gaps could also help to tackle this issues.

- The figures containing map mostly refer to a certain datum/projection. Still, this would need the specification of some units [km]; a scalebar would facilitate interpretation. - Commonly, table captions are displayed above a table, not below it.

---

## Author Comment (AC1) · 9 Aug 2017

**Response to the Short Comment 1 posted by Till Francke.**

*The manuscript by Avisse et al. describes an elaborated approach for retrieving the storage volumes of smaller reservoirs from remote sensing. As it relies exclusively on well-available optical and DEM data, it seems a valuable contribution for the monitoring of these storages in data-scarce regions. Since the authors also emphasise the general usefulness und transferability in this regard, I'd like to encourage them to share the required source code of the algorithm, which would match the spirit of publishing in an Open-Access-journal.*

[Figure]

We thank you for your constructive comments. We indeed think this method could be applied to other reservoirs in different parts of the world. We took your suggestion to make the code available; it will soon be possible to download it at a URL.

*Further minor suggestions: - Fig. 10 suggests that the methods tends to underestimate large volumes. Especially for Karama and Tanour there seems to be a upper limit, which the predictions of the method do not exceed. This is apparently not related to the complete filling of the reservoirs, as the ground observations confirm some dynamics within these phases. Is there any explanation to that?*

This is a good point. We think that this phenomenon may be caused by a very small variation of the water area for highest water levels. If the number of high-elevation pixels is small, the uncertainty on their corrected elevation (and thus the filling curve) can potentially affect the estimate of the maximum storage. This may indeed be a limitation of the method.

**We will add the following sentences in the revised version of the paper (p13, l12):** "For some reservoirs (i.e., Karama and Tanour), the method seems to have difficulties to predict highest storages. Indeed, if the number of high-elevation pixels is small, the uncertainty on their corrected elevation (and thus the filling curve) can potentially affect the estimate of the maximum storage. This may be a limitation of the method."

*- Table 2: The values of $eps_m(V)$ for Kafrein and King Talal differ surprisingly from the impression one gets in Fig 10: In the plot, Kafrein seems to be modelled much better than King Talal. Is there any explanation for this surprising impression?*

The selected error indicator does not represent well the quality of the results. This is due to the definition of $eps_m(V)$ itself. $eps_m(V)$ measures the ratio of errors compared to observed storages. Errors for Kafrein reservoir estimates are low, but the storage is

also sometimes very low. This explains why the relative error is high. For King Talal, observed storages are systematically higher that estimated ones, which may explain why $eps_m(V)$ tends to be so small.

**Proposed correction:** we would like to replace $eps_m$ with the NRMSE criteria defined as: $NRMSE = \frac{1}{Hist_{max}-Hist_{min}} \cdot \sqrt{\frac{1}{N}\sum_{i=1}^{N}\frac{(RS_i-Hist_i)^2}{N}}$ to give more weight to large errors than to smaller ones. The normalization is also done by considering (Vmax-Vmin) to consider the large range of storage variations. By using this criteria, errors vary between 10 % and 16 % for V and between 5 % and 30 % for H, and better match the impression we have when looking at the curves.

*- Specifying a relative error for H ($eps_m(H)$, Table 2, Fig.11) does not make sense to me: If H is water surface elevation, $eps_m$ will then depend on absolute altitude. Instead, water level (H - $H_{min}$) or absolute deviation ($H_{RS}$-$H_{HIST}$ - mean($H_{RS}H_{HIST}$)) should be used.*

Yes, you are right. We did not make it clear enough in the first version of the manuscript, but we did use (Hc-Hmin) to calculate the relative error. In the definition that we chose for the NRMSE, the normalization is now done using the difference between maximal and minimal values, so we do not have this issue anymore.

*- The choice of the regression used for reconstructing the H-A-relationship is not explained: According to Tab. 1, "Polynomial Regression" of different order and "Local Polynomial Regression" are used. Are they selected by best fit? The respective description (p. 10, ll. 15) is quite vague, especially concerning the 3-fold repetition of the process and the exclusion of outliers.*

**Proposed correction:** "To address the issue, a polynomial regression on observed land pixels (A > Ai, with Ai the area assumed as immersed during the satellite elevation retrieval) is used to build a "corrected elevation"-area relationship (A -> Hc(A)) **that**

**best fits the data (on a least-squares sense). Values of H greater than the $80^{th}$ or lower than the $20^{th}$ percentile are ignored** to filter potential errors and smooth the data. This step is executed three times – one for each DEM – and the better quality dataset **(i.e., the one with less dispersion and "anomalies" as defined above)** is kept. A few examples are shown in Fig. 7."

*- When discerning water surfaces, water bodies with macrophyte growth remain a serious challenge. It would be interesting to discuss if the presented approach for eliminating the SLC-data gaps could also help to tackle this issues.*

Macrophytes could indeed put a limitation to the detection of water bodies area. If there is no significant variation in the elevation, the 3D correction will not be as effective as with the Landsat N/A stripes to fill macrophyte-covered water areas. However, in that case the missing volume may not be significant if macrophytes do not cover a large part of the reservoir.

*- The figures containing map mostly refer to a certain datum/projection. Still, this would need the specification of some units [km]; a scalebar would facilitate interpretation.*

Thanks for this comment. We indeed forgot to specify that 1 unit equals 1 m. **We will add it in the legend of figures 3, 4 and 8.**

*- Commonly, table captions are displayed above a table, not below it.*

This is unfortunately out of our hands as we used the HESS Discussion template which defines the standards.

---

## Referee Comment (RC1) · Anonymous Referee #1 · 8 Sep 2017

The manuscript by Avisse et al. presents a novel approach to derive water level and water storage of small reservoirs based on optical images and DEMs. The methodology is a potentially valuable supplement to satellite altimetry, which traditionally is used for water level estimation and the results presented in the manuscript are promising. The paper is generally well written and well organized. I have some comments that are specified below.

General comments:

Section 2.2.2 that describes the elevation-area relationship needs to be more detailed possibly supplemented with illustrations in the same way as section 2.1 and figure 4 to enhance the understanding. E.g $r^q$ is not explained and step 3 is quite vague.

[Figure]

Please quantify or at least discuss the error on your estimated water levels/volumes.

specific comments:

P 2,l 10-16: The authors mention satellite altimetry. The authors could here mention the newer SAR missions CryoSat-2 and Sentinel-3, which have an along-track resolution of 300 m. The CryoSat-2 mission in SAR mode has demonstrated the potential of monitoring small lakes. Altimetry is not just limited to a few 100 lakes.

Eqn (3): As pointed out in the short comment Eqn (3) does not make sense and yes NRMSE is a good solution.

P 13, l8-10: "We can see coherent storage variations through the presence of drawdown-refill cycles, which means that the 2D enhancement and 3D reconstruction steps have improved the detection of water and helped to overcome the low Landsat repeat cycle of 16 days." I do not see this connection. How do you see that you improved the water detection when you are not comparing to anything?

P 14, l1-2. "Some of the differences between our estimates and measured data might then come from the inaccuracy regarding the data collection date". Is this because the in situ data are not daily?

Conclusion/Discussion: You could also mention the potential of Sentinel-1 and 2, which have a much higher resolution than Landsat.
* * *

---

## Author Comment (AC2) · 19 Sep 2017

*The manuscript by Avisse et al. presents a novel approach to derive water level and water storage of small reservoirs based on optical images and DEMs. The methodology is a potentially valuable supplement to satellite altimetry, which traditionally is used for water level estimation and the results presented in the manuscript are promising. The paper is generally well written and well organized. I have some comments that are specified below.*

Thank you for your time and your helpful comments. We indeed think that this method complements well current methods based on satellite altimetry to monitor more reservoirs, and to conduct studies for periods during which satellite altimetry did not

[Figure]

permit to estimate small reservoir storage variations.

*General comments:*

*Section 2.2.2 that describes the elevation-area relationship needs to be more detailed possibly supplemented with illustrations in the same way as section 2.1 and figure 4 to enhance the understanding. E.g $r^q$ is not explained and step 3 is quite vague.*

- We have added the figure below ("Fig. 1" refered to as Fig. X in the comments) in p10.

  **Proposed additional comment (p10, l6): "To illustrate the interest of this section, Figure X shows both $r^p$ and the relative elevation in the Kudnah reservoir. We can see that the two do not always match as we would expect – i.e. lowest pixels are not always the most frequently immersed, nor are the highest pixels the most rarely immersed. The immersion frequency ($r^p$) can actually be used to correct the elevation. The former, which is estimated from the results of the 2D classification enhancement, is indeed assumed more reliable than the original DEM. Hence,** each pixel's elevation $H^p$ is put in relation with the area $A^p$, defined as the cumulated area of all pixels $q$ **in the reservoir** for which $r^q \geq r^p$. The examples of Fig. 7 **confirm the observations made on Fig. X:** pixels' elevations are not always correlated with the number of times they are classified as water. To a certain extent **the difference** was expected from the DEM's low vertical precision, but some "anomalies" concerning the most often immersed pixels (i.e. lowest $A^p$) can be recurrent from one reservoir to another due to either a strong dispersion in elevation (see SRTM-X data in Fig. 7 (b)), or a flat elevation (see SRTM-C data in Fig. 7 (c))."...

- $r^q$ is given in Equation 1 (p 10). Indeed, by replacing $p$ with $q$ (for all other pixels in the reservoirs), we get: $r^q = \frac{N^q_{water}}{N^q_{water}+N^q_{land}}$ where $N^q_{water}$ is the number of times

the given pixel $q$ is counted as Water, and $N^q_{land}$ the number of times it is counted as Land. Images where the pixel $q$ is classified as Unknown are ignored.

**Proposed correction (p10, l6):** "Each pixel's elevation $H^p$ is put in relation with the area $A^p$, defined as the cumulated area of all pixels $q$ **in the reservoir** for which $r^q \geq r^p$."

- We realised step 3 was not clear enough with the first short comment published by T. Francke in the open discussion. We thus added several corrections to detail each part of step 3.

**Proposed correction (p10, l15):** "To address the issue, a polynomial regression on observed land pixels ($A > A_i$, with $A_i$ the area assumed as immersed during the satellite elevation retrieval) is used to build a "corrected elevation"-area relationship ($A \rightarrow H_c(A)$) **that best fits the data (on a least-squares sense). Values of $H$ greater than the 80$^{th}$ or lower than the 20$^{th}$ percentile are ignored** to filter potential errors and smooth the data. This step is executed three times – one for each DEM – and the better quality dataset **(i.e. the one with less dispersion and fewer "anomalies" as defined above)** is kept. Examples are shown in Fig. 7."

*Please quantify or at least discuss the error on your estimated water levels/volumes.*

Identifying the cause to the errors of the method is indeed fundamental for further research on reservoir monitoring using DEM. The errors associated with the elevation-area relationships must be found in the DEMs' low vertical precision: up to 15 m for 90 % of the data for SRTM-C, 6 m for 90 % of the data for SRTM-X, and a standard deviation of 8.68 m for the ASTER dataset (p9, l13-15). Errors on the corrected elevation are difficult to quantify from the curves presented in Fig. 7, as we do not know the error that should actually be expected for each reservoir's location. All we can derive from the area-elevation relationships is that "pixels' elevation is not always correlated with

the number of times they are classified as water [because of the] DEM's low vertical precision" (p10, l7-8).

This is the reason why we can only quantify changes in the reconstruction of water bodies area (see Table 3) and measure the error on water levels and volumes after all correction steps are completed (see Table 2). We cannot decompose the error and associate each part to a certain step without making unfunded assumptions. The discussion on the error of the results is given on p13, l11 to p16, l11.

*specific comments:*

*P 2, l 10-16: The authors mention satellite altimetry. The authors could here mention the newer SAR missions CryoSat-2 and Sentinel-3, which have an along-track resolution of 300 m. The CryoSat-2 mission in SAR mode has demonstrated the potential of monitoring small lakes. Altimetry is not just limited to a few 100 lakes.*

Yes, you are right, we forgot to mention these satellites. They indeed provide large improvements in terms of resolution. The reference to the few hundred large lakes and reservoirs on the planet actually comes from a statement made by Gao et al. (2012) – also mentioned in Zhang et al. (2014). But the more recent satellites mentioned by the reviewer clearly increased the potential monitoring of water level to thousands of lakes and reservoirs (Crétaux et al. 2016).

However, Cryosat was initially designed to monitor the thickness of Arctic ice, and SAR and SARin modes are only available over a limited surface of the Earth (mainly high latitudes and mountain glaciers, see European Space Agency (2012)). Its revisit cycle of 369 days also impedes the monitoring of small reservoirs on a monthly basis. Moreover, the inter-track of ∼7 km and ∼52 km at the equator for CryoSat-2 and the two Sentinel-3 satellites respectively (Donlon et al., 2012; Crétaux et al., 2016) still make many small reservoirs out of the trajectory of the nadir-viewing sensors onboard.

[Figure]

It can be noted that Crétaux (2016) also summarised errors on measurements estimates for a set of lakes and reservoirs of various size and location to highlight the fact that many elevation measurements over a water body are necessary to get a precise estimate. Zhang et al. (2014) estimate that a distance of 10 km is necessary to take enough measurements to get a precise elevation assessment with "older" satellite altimeters (i.e. with an along-track path resolution of 1 km). By considering the same ratio of measurements per distance crossed, the width of reservoir bodies observed by satellites like Sentinel-3 (i.e., resolution of 300 m) still needs to reach 3 km, which is larger than most reservoirs considered in our study – assuming that the satellite passed at the right location over them.

Then, for these reasons and as attested to by research studies that used these satellites, the focus has been made on reservoirs larger than 100 km$^2$ (Crétaux et al., 2016; Jiang et al., 2017) or rivers that stretch over several hundred kilometres (Villadsen et al., 2015).

**Proposed correction (p2, l11):** "They have a high vertical accuracy with root mean square errors on the order of centimetres to tens of centimetres depending on the altimeter and the size of the water body (Calmant et al., 2008; **Crétaux, 2016**). Yet, **the above mentioned** sensors are affected by important drawbacks, including nadir viewing, narrow swath, coarse cross-track spacing (a few hundred kilometres), long along-track path length (about 1 km), large elevation differences around some water areas, that impede their application to more than a few hundred large lakes and reservoirs on the planet (i.e. area > 100 km$^2$ and width > 500 m) (Crétaux and Birkett, 2006; Alsdorf et al., 2007; Gao et al., 2012; **Zhang et al., 2014). More recent satellites such as Cryostat-2 or Sentinel-3 present significant improvements in terms of along-track resolution ($\sim$300 m). However, their respective inter-track of 7 km and 52 km (Donlon et al., 2012; Crétaux et al., 2016) still place many reservoirs out of the trajectory of their nadir-viewing sensors onboard. The small inter-track of Cryosat is also realised at the expense of a long revisit cycle (369 days) that**

**impedes any monitoring of small reservoirs on a monthly basis**."

**Proposed additional precision in the introduction (p1, l4):** "This paper proposes a novel approach using Landsat imagery and Digital Elevation Models (DEM) to retrieve information on storage variations in **any** inaccessible region."

**Proposed additional precision for the conclusion (p19, l9):** "For all "uncovered" small or large reservoirs, the uses of datasets available over the whole continental surface **make this method a valuable complement to satellite altimetry to increase the number of reservoirs observable anywhere in the world.** The thresholds dynamically defined for both the 2D enhancement and the 3D reconstruction **also** make the method potentially suitable to monitor reservoirs in **truly** inaccessible areas."

*Eqn (3): As pointed out in the short comment Eqn (3) does not make sense and yes NRMSE is a good solution.*

We thank you for this response to the first interactive comment. We consider your advice and keep the NRMSE criteria to quantify the error of our estimates.

*P 13, l8-10: "We can see coherent storage variations through the presence of drawdown-refill cycles, which means that the 2D enhancement and 3D reconstruction steps have improved the detection of water and helped to overcome the low Landsat repeat cycle of 16 days." I do not see this connection. How do you see that you improved the water detection when you are not comparing to anything?*

Yes indeed, this argument alone does not make sense. Thank you for noticing the mistake. Changes in water bodies' area obtained with similar tools – but without the 2D enhancement and 3D reconstruction – are available in the supporting information (Figure S3) of the study conducted by Müller et al. (2016). Improvements in the detection of annual drawdown-refill cycles are particularly clear for Sahwat al-Khadr and Roum

dams.

**Proposed correction (p13, l8):** "**By qualitatively comparing our results to those obtained by Müller et al. (2016) (monitoring of Syrian reservoirs using Landsat 7 datasets but before the 2D and 3D corrections),** we can see **more** coherent storage variations through the presence of **annual** drawdown-refill cycles – **particularly for Sahwat al-Khadr and Roum. It** means that the 2D enhancement and 3D reconstruction steps have improved the detection of water and helped to overcome the low Landsat repeat cycle of 16 days."

*P 14, l1-2. "Some of the differences between our estimates and measured data might then come from the inaccuracy regarding the data collection date". Is this because the in situ data are not daily?*

Yes indeed.

**Proposed correction (p13, l11-12):** "Reservoirs managed by Jordan are used to validate the method by comparing our remote sensing estimates of elevation and storage with **monthly** *in situ* measurements conducted by the Jordan Valley Authority (JVA)."

*Conclusion/Discussion: You could also mention the potential of Sentinel-1 and 2, which have a much higher resolution than Landsat.*

Yes indeed, thank you for mentioning these two satellites. Actually, as explained p2, l28, we chose to not detail SAR sensors as they have "been less used due to the difficulty to get consistent results, as the required condition of a significantly lower phase coherence of water areas than of the surrounding land surface is not always met with orbital repeat cycles of more than a few days, or with wind or rain (Alsdorf et al., 2007; Eilander et al., 2014)".

**Proposed correction (p2, l27):** "Water surface areas are commonly determined from optical satellite imagery such as MODerate Resolution Imaging Spectroradiometer (MODIS) and Landsat products (Xiao et al., 2006; Gao et al., 2012), or Synthetic Aperture Radar (SAR) sensors (e.g., RADARSAT, JERS-1, ERS **or Sentinel-1**) (Annor et al., 2009; Duan and Bastiaanssen, 2013; **Amitrano, 2014**)."

With regard to Sentinel 2, new references are added in the introduction.

**Proposed correction (p3, l2): "The potential of the recent two Sentinel-2 satellites can also be mentioned for the study of recent or future years. Launched in June 2015 (Sentinel-2A) and March 2017 (Sentinel-2B), they provide spectral bands at a resolution of 10 m for visible and NIR bands, and at 20 m for SWIR bands. They also have a repeat cycle of 5 days by combining the 2 (European Space Agency, 2013; Yang et al., 2017)."**

References:

Alsdorf, D. E., Rodríguez, E., and Lettenmaier, D. P.: Measuring surface water from space, Reviews of Geophysics, 45, doi:10.1029/2006RG000197, rG2002, 2007.

Amitrano, D., Martino, G. D., Iodice, A., Mitidieri, F., Papa, M. N., Riccio, D., and Ruello, G.: Sentinel-1 for Monitoring Reservoirs: A Performance Analysis, Remote Sensing, 6, 10676 – 10693, doi:10:3390/rs61110676, 2014.

Calmant, S., Seyler, F., and Crétaux, J. F.: Monitoring Continental Surface Waters by Satellite Altimetry, Surveys in Geophysics, 29, 247–269, doi:10.1007/s10712-008-9051-1, 2008.

Crétaux, J.-F. and Birkett, C.: Lake studies from satellite radar altimetry, Comptes Rendus Geoscience, 338, 1098–1112, doi:10.1016/j.crte.2006.08.002, 2006.

Crétaux, J.-F., Abarca-del Río, R., Bergé-Nguyen, M., Arsen, A., Drolon, V., Clos, G., and Maisongrande, P.: Lake Volume Monitoring from Space, Surveys in Geophysics,

37, 269 – 305, doi:10:1007/s10712-016-9362-6, 2016.

Donlon, C., Berruti, B., Buongiorno, A., Ferreira, M.-H., Féménias, P., Frerick, J., Goryl, P., Klein, U., Laur, H., Mavrocordatos, C., Nieke, J., Rebhan, H., Seitz, B., Stroede, J., and Sciarra, R.: The Global Monitoring for Environment and Security (GMES) Sentinel-3 mission, Remote Sensing of Environment, 120, 37 – 57, doi:10.1016/j:rse:2011:07:024, the Sentinel Missions – New Opportunities for Science, 2012.

Eilander, D., Annor, F. O., Iannini, L., and van de Giesen, N.: Remotely Sensed Monitoring of Small Reservoir Dynamics: A Bayesian Approach, Remote Sensing, 6, 1191 – 1210, doi:10:3390/rs6021191, 2014.

European Space Agency: CryoSat Product Handbook, available at: https://earth.esa.int/documents/10174/125272/CryoSat_Product_Handbook, 2012.

European Space Agency: Sentinel-2 Mission Details, available at: https://earth.esa.int/web/guest/missions/esa-operational-eo-missions/sentinel-2, 2013.

Gao, H., Birkett, C., and Lettenmaier, D. P.: Global monitoring of large reservoir storage from satellite remote sensing, Water Resources Research, 48, doi:10.1029/2012WR012063, w09504, 2012.

Jiang, L., Nielsen, K., Andersen, O. B., and Bauer-Gottwein, P.: Monitoring recent lake level variations on the Tibetan Plateau using CryoSat-2 SARIn mode data, Journal of Hydrology, 544, 109 – 124, doi:10:1016/j:jhydrol:2016:11:024, 2017.

Müller, M. F., Yoon, J., Gorelick, S. M., Avisse, N., and Tilmant, A.: Impact of the Syrian refugee crisis on land use and transboundary freshwater resources, Proceedings of the National Academy of Sciences, doi:10.1073/pnas.1614342113, 2016.

Villadsen, H., Andersen, O. B., Stenseng, L., Nielsen, K., and Knudsen, P.: CryoSat-2 altimetry for river level monitoring – Evaluation in the Ganges-Brahmaputra River basin,

Remote Sensing of Environment, 168, 80 – 89, doi:10:1016/j:rse:2015:05:025, 2015.

Yang, X., Zhao, S., Qin, X., Zhao, N., and Liang, L.: Mapping of Urban Surface Water Bodies from Sentinel-2 MSI Imagery at 10 m Resolution via NDWI-Based Image Sharpening, Remote Sensing, 9, doi:10:3390/rs9060596, 2017.

Zhang, S., Gao, H., and Naz, B. S.: Monitoring reservoir storage in South Asia from multisatellite remote sensing, Water Resources Research, 50, 8927–8943, doi:10.1002/2014WR015829, 2014.

[Figure]

[Figure]

[Figure]

**Fig. 1.** (a) Relative immersion frequency (r^p, from the 2D classes) and (b) relative elevation (from the DEMs, in terms of percentile) in the Kudnah reservoir.

---

## Referee Comment (RC2) · W. Gumindoga (Referee) · 26 Sep 2017

I read with enthusiasm the paper by Nicolas Avisse et al on Monitoring small reservoirs storage from satellite remote sensing in inaccessible areas. The approach to use satellite data (Landsat imagery and Digital Elevation Models (DEM)) to retrieve information on storage variations in ungauged and inaccessible areas is welcome for improved water resource management. A question arises for the Fmask function for distinguishing land and water areas and producing a probability mask for clouds. What specific criteria was used to manually remove images that are almost entirely covered by clouds or with obvious large errors in water bodies detection?
 What specific quality control measures did the authors take to remain with 245 images per location?
 The authors can do justice by quantifying the uncertainty in the

[Figure]

Fmask method.
 In Section 2.1.3, how realistic is to define automatically the threshold for optimally distinguishing water bodies from clouds using the MNDWI technique?
 Authors can also justify the selection of Landsat 7 images over the more recent Landsat 8, which do not have stripes after all.
 Section 3.1 what do the authors mean by saying "...some of the differences between our estimates and measured data might then come from the inaccuracy regarding the data collection date."
 The authors need to improve on the equality of the maps by improving on some map fundamentals/basics such as north arrow, legend and scale.
 Why not validating the elevation-area relationships with some established/measured rating curves

---

## Author Comment (AC3) · 10 Oct 2017

*Conclusion/Discussion: You could also mention the potential of Sentinel-1 and 2, which have a much higher resolution than Landsat.*

**Proposed additional correction for the conclusion (p19, l15)**: "**The recent two Sentinel-2 satellites also promise a great improvement of the method for post-2015 studies, as they have spatial and temporal resolutions finer than Landsat (up to 10 m and 5 days). Combining Landsat and Sentinel-2 satellites would then reduce the already short revisit cycle of water bodies and would provide near real-time updates on water bodies storage**".

Furthermore, the algorithms used in the paper automatically detect water bodies,

define the water areas retrieval parameters, build filling curves and assess reservoir storage. **Such algorithmic tools can then be dynamically updated with each new image from Sentinel-2 and Landsat satellites, giving the model the potential to learn by itself and correct previous storage estimates while generating new ones. This approach is somehow comparable to the continuous change detection proposed by Zhu et al. (2014)**."

References:

Zhu, Z. and Woodcock, C. E.: Continuous change detection and classification of land cover using all available Landsat data, Remote Sensing of Environment, 144, 152 - 171, doi:/10:1016/j:rse:2014:01:011, 2014.

---

## Author Comment (AC4) · 10 Oct 2017

*I read with enthusiasm the paper by Nicolas Avisse et al on Monitoring small reservoirs storage from satellite remote sensing in inaccessible areas. The approach to use satellite data (Landsat imagery and Digital Elevation Models (DEM)) to retrieve information on storage variations in ungauged and inaccessible areas is welcome for improved water resource management.*

Thank you for your interest and for taking time commenting our paper.

*A question arises for the Fmask function for distinguishing land and water areas and producing a probability mask for clouds. What specific criteria was used to manually*

[Figure]

*remove images that are almost entirely covered by clouds or with obvious large errors in water bodies detection? What specific quality control measures did the authors take to remain with 245 images per location? The authors can do justice by quantifying the uncertainty in the Fmask method.*

The analysis conducted to "manually remove images that are almost entirely covered by clouds or with obvious large errors in water bodies detection" (p4, l12) is a rough observation of Fmask classification results (mainly for categories 'clouds' and 'water' as mentioned above). The quality control is a visual comparison between these classification results and original images (SWIR-R-G for instance). Zhu et al. (2012) evaluate a cloud overall accuracy of 96.41 %, but it depends a lot on the satellite, location and time: Zhu et al. (2015) estimate an overall accuracy (i.e., for all classes) varying between 24 % and 89 %, for instance, depending on the Landsat 8 image chosen. Also, according to our study (p6, l1-2): "on average, 24.1 % of reservoirs' pixels are misclassified as land, 8.1 % are covered with clouds or cloud shadows, and 8.6 % are in 'N/A' areas".

In fact, the objective of this step is not to precisely detect clouds or water areas. We just need a first rough selection of images and remove those that could affect the next statistical analyses (statistical correction of elevation and 3D reconstruction through hidden areas). For instance, if Fmask detects clouds over the whole image, then it cannot be used in the next steps. Similarly, if Fmask classifies half an image as water, it is obviously a misdetection from the algorithm. By removing such images, we went from 300 to 245 images per location.

**Proposed correction (p5, l10):** "The algorithm was originally designed to separate potential cloud pixels from clear sky pixels on Landsat images using empirical thresholds on NDVI and the near-infrared band, **with an overall accuracy of 96.41 % (Zhu et al., 2012)**."

*In Section 2.1.3, how realistic is to define automatically the threshold for optimally distinguishing water bodies from clouds using the MNDWI technique?*

As the referee W. Gumindoga rightly points out, an automatic classification with MNDWI or NDVI (or any other criteria) will not give as good results as if we chose a specific criterion for each reservoir at each time. A trade-off is indeed required between the time to spend on the detection and the quality of the results.

As explained in the introduction (p3, l3-13), various methods have been applied to detect water areas. The most basic ones rely on a predefined NDVI or MNDWI threshold, which is problematic for a multi-temporal analysis (Liu et al., 2012). Coltin et al. (2016) give an inventory of other indices generally used for detecting water, and advocate the implementation of automatic thresholds as they develop a supervised learning approach. Other methods rely on an automatic unsupervised classification (Wang et al., 2008; Gao et al., 2012). In our paper, we choose to automatically define a threshold for each image. Our protocol has actually the advantage of being entirely automatic (no further association between class and type of land use for instance). This approach is very fast, no selection of reservoir approximate location is required, and, as mentioned in the conclusion, it could "provide near real-time updates on water bodies storage".

**Proposed additional reference (p3, l5-7)**: "But determining an adequate value for a multi-temporal analysis can be challenging because such a threshold is known to be case-dependent (Liu et al., 2012; **Coltin et al., 2016**)."

**Other additional correction (p3, l8):** "To address these issues, decision tree defined thresholds have successfully been applied with various vegetation indices (e.g., Xiao et al., 2006; Islam et al., 2010; Yan et al., 2010), but remain case-dependent. **Coltin et al. (2016) have then advocated the implementation of automatic thresholds as they developed a supervised learning approach to improve flood mapping**."

*Authors can also justify the selection of Landsat 7 images over the more recent Landsat*

*8, which do not have stripes after all.*

Yes you are right, Landsat 8 do have the advantage of not having stripes. We actually used all kinds of Landsat images including Landsat 8: "about 300 Landsat 4, 5, 7 and 8 images for each scene [...] are downloaded from the [USGS website]" (p5, l4-7). The goal is to use all Landsat images available to analyse changes in reservoir storage over long periods of time (ideally several decades), and Landsat 8 images are only available from February 2013.

*Section 3.1 what do the authors mean by saying "...some of the differences between our estimates and measured data might then come from the inaccuracy regarding the data collection date."*

As pointed out by the Anonymous Referee 1, we did not mention that the "*in situ* measurements conducted by the Jordan Valley Authority" are monthly. Then, as these measurements are not automatically recorded, we do not know exactly on which day they were collected, and if they are always collected the same day in the month. Such uncertainty may change the difference between our estimates and measured data.

**We proposed the following correction when answering the Referee 1 comment (p13, l11-12):** "Reservoirs managed by Jordan are used to validate the method by comparing our remote sensing estimates of elevation and storage with **monthly** *in situ* measurements conducted by the Jordan Valley Authority (JVA)."

**Proposed additional correction (p14, l1-2):** "**Because no information is available regarding the data collection date**, some of the differences between our estimates and measured data might then come from **this lack of metadata**."

*The authors need to improve on the equality of the maps by improving on some map fundamentals/basics such as north arrow, legend and scale.*

As pointed out by T. Francke in the Short Comment 1, we indeed forgot to specify that 1 unit equals 1 m for the scale.

**We have updated the legend of figures 3, 4 and 8. We have also added North and East indices to the legend to match common representation of satellite images (e.g., Frappart et al., 2006; Gao et al., 2012; Zhang et al., 2014; Crétaux, 2015). The title "Inundation frequency" is now also added next to the colorbar in Figure 3.**

*Why not validating the elevation-area relationships with some established/measured rating curves*

We unfortunately do not have such relationships for Syrian nor Jordanian reservoirs. As the observed elevation and volume for Jordanian reservoirs do not represent the whole range of possibilities, and because few elevation measurements were available ($15 < N < 35$, see p16, l10) and not necessarily conducted at the same time as storage measurements, the relationships could not be retrieved with precision.

References:

**Coltin, B., McMichael, S., Smith, T., and Fong, T.: Automatic boosted flood mapping from satellite data, International Journal of Remote Sensing, 37, 993 – 1015, doi:/10:1080/01431161:2016:1145366, 2016.**

Crétaux, J.-F., Biancamaria, S., Arsen, A., Bergé-Nguyen, M., and Becker, M.: Global surveys of reservoirs and lakes from satellites and regional application to the Syrdarya river basin, Environ. Res. Lett., 10, doi:10.1088/1748-9326/10/1/015002, 2015.

Frappart, F., Minh, K. D., L'Hermitte, J., Cazenave, A., Ramillien, G., Le Toan, T., and Mognard-Campbell, N.: Water volume change in the lower Mekong from satellite altimetry and imagery data, Geophysical Journal International, 167, 570–584,

doi:10.1111/j.1365-246X.2006.03184.x, 2006.

Gao, H., Birkett, C., and Lettenmaier, D. P.: Global monitoring of large reservoir storage from satellite remote sensing, Water Resources Research, 48, doi:10.1029/2012WR012063, w09504, 2012.

Islam, A., Bala, S., and Haque, M.: Flood inundation map of Bangladesh using MODIS time-series images, Journal of Flood Risk Management, 3, 210–222, doi:10.1111/j.1753-318X.2010.01074.x, 2010.

Liu, Y., Song, P., Peng, J., and Ye, C.: A physical explanation of the variation in threshold for delineating terrestrial water surfaces from multi-temporal images: effects of radiometric correction, International Journal of Remote Sensing, 33, 5862–5875, doi:10.1080/01431161.2012.675452, 2012.

Wang, Y., Sun, G., Liao, M., and Gong, J.: Using MODIS images to examine the surface extents and variations derived from the DEM and laser altimeter data in the Danjiangkou Reservoir, China, International Journal of Remote Sensing, 29, 293–311, doi:10.1080/01431160701253311, 2008.

Xiao, X., Boles, S., Frolking, S., Li, C., Babu, J. Y., Salas, W., and III, B. M.: Mapping paddy rice agriculture in South and Southeast Asia using multi-temporal MODIS images, Remote Sensing of Environment, 100, 95–113, doi:10.1016/j.rse.2005.10.004, 2006.

Yan, Y.-E., Ouyang, Z.-T., Guo, H.-Q., Jin, S.-S., and Zhao, B.: Detecting the spatiotemporal changes of tidal flood in the estuarine wetland by using MODIS time series data, Journal of Hydrology, 384, 156–163, doi:10.1016/j.jhydrol.2010.01.019, 2010.

Zhang, S., Gao, H., and Naz, B. S.: Monitoring reservoir storage in South Asia from multisatellite remote sensing,Water Resources Research, 50, 8927–8943, doi:10.1002/2014WR015829, 2014.

Zhu, Z. and Woodcock, C. E.: Object-based cloud and cloud shadow detection in Landsat imagery, Remote Sensing of Environment, 118, 83–94, doi:10.1016/j.rse.2011.10.028, 2012.

Zhu, Z., Wang, S., and Woodcock, C. E.: Improvement and expansion of the Fmask algorithm: cloud, cloud shadow, and snow detection for Landsats 4-7, 8, and Sentinel 2 images, Remote Sensing of Environment, 159, 269–277, doi:10.1016/j.rse.2014.12.014, 2015.
* * *

---

## Author Response (AR1)

Nicolas Avisse
Université Laval
1045 Rue de la Médecine
Québec, QC
G1V 0A6
Canada

November 07, 2017

Discussion paper reference: **hess-2017-373**

Dear Dr Mazvimavi,

Please find in the following the revised version of our manuscript "Monitoring small reservoirs storage from satellite remote sensing in inaccessible areas". We really appreciate the feedbacks provided by T. Francke, W. Gumindoga and an anonymous referee. Their comments and suggestions are first presented with our responses and proposed corrections to the manuscript, and a marked-up manuscript version is then given with all modifications.

Sincerely,

Nicolas Avisse
Corresponding Author
nicolas.avisse@gmail.com

**Response to the Short Comment 1 posted by Till Francke.**

*The manuscript by Avisse et al. describes an elaborated approach for retrieving the storage volumes of smaller reservoirs from remote sensing. As it relies exclusively on well-available optical and DEM data, it seems a valuable contribution for the monitoring of these storages in data-scarce regions. Since the authors also emphasise the general usefulness und transferability in this regard, I'd like to encourage them to share the required source code of the algorithm, which would match the spirit of publishing in an Open-Access-journal.*

We thank you for your constructive comments. We indeed think this method could be applied to other reservoirs in different parts of the world.

**We took your suggestion to make the code available (see *code and data availability*); it is now possible to download it at https://drive.google.com/open?id=0B54cRCK06X-9RUdqaTZmWkdsOXc.**

*Further minor suggestions: - Fig. 10 suggests that the methods tends to underestimate large volumes. Especially for Karama and Tanour there seems to be a upper limit, which the predictions of the method do not exceed. This is apparently not related to the complete filling of the reservoirs, as the ground observations confirm some dynamics within these phases. Is there any explanation to that?*

This is a good point. We think that this phenomenon may be caused by a very small variation of the water area for highest water levels. If the number of high-elevation pixels is small, the uncertainty on their corrected elevation (and thus the filling curve) can potentially affect the estimate of the maximum storage. This may indeed be a limitation of the method.

**We added the following sentences in the revised version of the paper (p14, l13):** "**For some reservoirs (i.e., Karama and Tanour), the method seems to have difficulties to predict highest storages. Indeed, if the number of high-elevation pixels is small, the uncertainty on their corrected elevation (and thus the filling curve) can potentially affect the estimate of the maximum storage. This may be a limitation of the method.**"

*- Table 2: The values of eps_m(V) for Kafrein and King Talal differ surprisingly from the impression one gets in Fig 10: In the plot, Kafrein seems to be modelled much better than King Talal. Is there any explanation for this surprising impression?*

The selected error indicator does not represent well the quality of the results. This is due to the definition of $eps_m(V)$ itself. $eps_m(V)$ measures the ratio of errors compared to observed storages. Errors for Kafrein reservoir estimates are low, but the storage is also sometimes very low. This explains why the relative error is high. For King Talal, observed storages are systematically higher than estimated ones, which may explain why $eps_m(V)$ tends to be so small.

**Proposed correction: We replaced eps_m with the NRMSE criteria (p17, l5) defined as:**

$$NRMSE = \frac{\sqrt{\frac{1}{N}\sum_{i=1}^{N}\frac{(RS_i - Hist_i)^2}{N}}}{Hist_{max} - Hist_{min}}$$ **to give more weight to large errors than to smaller ones. The normalisation is also done by considering ($V_{max}$-$V_{min}$) to consider the large amplitude of storage variations. By using this criteria, errors vary between 10 % and 16 % for V and between 5 % and 30 % for H, and better match the impression we have when looking at the curves.**

*- Specifying a relative error for H (eps_m(H), Table 2, Fig.11) does not make sense to me: If H is water surface elevation, eps_m will then depend on absolute altitude. Instead, water level (H - H_min) or absolute deviation (H_RS-H_HIST - mean(H_RSH_HIST)) should be used.*

Yes, you are right. We did not make it clear enough in the first version of the manuscript, but we did use ($H_c$-$H_{min}$) to calculate the relative error. In the definition that we chose for the NRMSE, the normalisation is now done using the difference between maximal and minimal values, so we do not have this issue anymore.

*- The choice of the regression used for reconstructing the H-A-relationship is not explained: According to Tab. 1, "Polynomial Regression" of different order and "Local Polynomial Regression" are used. Are they selected by best fit? The respective description (p. 10, ll. 15) is quite vague, especially concerning the 3-fold repetition of the process and the exclusion of outliers.*

**Proposed correction (p11, l4):** "To address the issue, a polynomial regression on observed land pixels (A > $A_i$, with $A_i$ the area assumed as immersed during the satellite elevation retrieval) is used to build a "corrected elevation"-area relationship (A -> $H_c$(A)) that **best fits the data (on a least-squares sense). Values** of H **greater than the 80th or lower than the 20th percentile are ignored to filter potential errors and smooth the data.** This step is executed three times – one for each DEM – and the better quality dataset **(i.e., the one with less dispersion and "anomalies" as defined above)** is kept. Examples are shown in Fig. 8."

*- When discerning water surfaces, water bodies with macrophyte growth remain a serious challenge. It would be interesting to discuss if the presented approach for eliminating the SLC-data gaps could also help to tackle this issues.*

Macrophytes could indeed put a limitation to the detection of water bodies area. If there is no significant variation in the elevation, the 3D correction will not be as effective as with the Landsat N/A stripes to fill macrophyte-covered water areas. However, in that case the missing volume may not be significant if macrophytes do not cover a large part of the reservoir.

*- The figures containing map mostly refer to a certain datum/projection. Still, this would need the specification of some units [km]; a scalebar would facilitate interpretation.*

Thanks for this comment. We indeed forgot to specify that 1 unit equals 1 m.

**It has been added in the legend of figures 1, 3, 4, 7 and 9.**

*- Commonly, table captions are displayed above a table, not below it.*

This is unfortunately out of our hands as we used the HESS Discussion template which defines the standards.

**Response to the Referee Comment 1 posted by Anonymous Referee 1:**

*The manuscript by Avisse et al. presents a novel approach to derive water level and water storage of small reservoirs based on optical images and DEMs. The methodology is a potentially valuable supplement to satellite altimetry, which traditionally is used for water level estimation and the results presented in the manuscript are promising. The paper is generally well written and well organized. I have some comments that are specified below.*

Thank you for your time and your helpful comments. We indeed think that this method complements well current methods based on satellite altimetry to monitor more reservoirs, and to conduct studies for periods during which satellite altimetry did not permit to estimate small reservoir storage variations.

*General comments:*

*Section 2.2.2 that describes the elevation-area relationship needs to be more detailed possibly supplemented with illustrations in the same way as section 2.1 and figure 4 to enhance the understanding. E.g r^q is not explained and step 3 is quite vague.*

**- We have added the following figure (p11):**

[Figure]

"**Figure 7: (a) Relative non-immersion frequency (1 - r$^p$, from the 2D classes) and (b) elevation (from the DEMs, in terms of decile) in the Kudnah reservoir. Coordinates are expressed in CRS WGS 84 / UTM zone 36N, in which 1 unit equals 1 m.**"

**Proposed additional comment (p10, l9): "To illustrate the interest of this section, Figure 7 shows both r$^p$ and the relative elevation in the Kudnah reservoir. We can see that the two do not always match as we would expect – i.e. lowest pixels are not always the most frequently immersed, nor are the highest**

**pixels the most rarely immersed. The immersion frequency ($r^p$) can actually be used to correct the elevation. The former, which is estimated from the results of the 2D classification enhancement, is indeed assumed more reliable than the original DEM. Hence,** each pixel's elevation $H^p$ is put in relation with the area $A^p$, defined as the cumulated area of all pixels q **in the reservoir** for which $r^q \geq r^p$. The examples of Fig. 8 **confirm the observations made on Fig. 7: pixels' elevations are** not always correlated with the number of times they are classified as water. To a certain extent **the difference** was expected from the DEM's low vertical precision, but some "anomalies" concerning the most often immersed pixels (i.e. lowest $A^p$) can be recurrent from one reservoir to another due to either a strong dispersion in elevation (see SRTM-X data in Fig. 8 (b)), or a flat elevation (see SRTM-C data in Fig. 8 (c))."

- $r^q$ is given in Equation 1 (p 10). Indeed, by replacing p with q (for all other pixels in the reservoirs), we get:

$$r^q = \frac{N^q_{water}}{N^q_{water} + N^q_{land}}$$

where N(q, water) is the number of times the given pixel q is counted as Water, and N(q, land) the number of times it is counted as Land. Images where the pixel q is classified as Unknown are ignored.

**Proposed correction (p10, l13):** "Each pixel's elevation $H^p$ is put in relation with the area $A^p$, defined as the cumulated area of all pixels q **in the reservoir** for which $r^q \geq r^p$."

- We realised step 3 was not clear enough with the first short comment published by T. Francke in the open discussion. We thus added several corrections to detail each part of step 3 (see above also).

**Proposed correction (p11, l4):** "To address the issue, a polynomial regression on observed land pixels (A > $A_i$, with $A_i$ the area assumed as immersed during the satellite elevation retrieval) is used to build a "corrected elevation"-area relationship (A -> $H_c(A)$) that **best fits the data (on a least-squares sense). Values** of H **greater than the 80$^{th}$ or lower than the 20$^{th}$ percentile are ignored to filter potential errors and smooth the data.** This step is executed three times – one for each DEM – and the better quality dataset **(i.e., the one with less dispersion and "anomalies" as defined above)** is kept. Examples are shown in Fig. 8."

*Please quantify or at least discuss the error on your estimated water levels/volumes.*

Identifying the cause to the errors of the method is indeed fundamental for further research on reservoir monitoring using DEM. The errors associated with the elevation-area relationships must be found in the DEMs' low vertical precision: up to 15 m for 90 % of the data for SRTM-C, 6 m for 90 % of the data for SRTM-X, and a standard deviation of 8.68 m for the ASTER dataset (p9, l16). Errors on the corrected elevation are difficult to quantify from the curves presented in Fig. 7, as we do not know the error that should actually be expected for each reservoir's location. All we can derive from the area-elevation relationships is that "pixels' elevations are not always correlated with the number of times they are classified as water [because of the] DEM's low vertical precision" (p10, l15-17).

This is the reason why we can only quantify changes in the reconstruction of water bodies area (see Table 3) and measure the error on water levels and volumes after all correction steps are completed (see Table 2). We cannot decompose the error and associate each part to a certain step without making unfunded assumptions. The discussion on the error of the results is given on p14, l5 to p17, l14.

*specific comments:*

*P 2, l 10-16: The authors mention satellite altimetry. The authors could here mention the newer SAR missions CryoSat-2 and Sentinel-3, which have an along-track resolution of 300 m. The CryoSat-2 mission in SAR mode has demonstrated the potential of monitoring small lakes. Altimetry is not just limited to a few 100 lakes.*

Yes, you are right, we forgot to mention these satellites. They indeed provide large improvements in terms of resolution. The reference to the few hundred large lakes and reservoirs on the planet actually comes from a statement made by Gao et al. (2012) – also mentioned in Zhang et al. (2014). But the more recent satellites mentioned by the reviewer clearly increased the potential monitoring of water level to thousands of lakes and reservoirs (Crétaux et al. 2016).

However, Cryosat was initially designed to monitor the thickness of Arctic ice, and SAR and SARin modes are only available over a limited surface of the Earth (mainly high latitudes and mountain glaciers, see European Space Agency (2012)). Its revisit cycle of 369 days also impedes the monitoring of small reservoirs on a monthly basis. Moreover, the inter-track of ≈7 km and ≈52 km at the equator for CryoSat-2 and the two Sentinel-3 satellites respectively (Donlon et al., 2012; Crétaux et al., 2016) still make many small reservoirs out of the trajectory of the nadir-viewing sensors onboard.

It can be noted that Crétaux (2016) also summarised errors on measurements estimates for a set of lakes and reservoirs of various size and location to highlight the fact that many elevation measurements over a water body are necessary to get a precise estimate. Zhang et al. (2014) estimate that a distance of 10 km is necessary to take enough measurements to get a precise elevation assessment with "older" satellite altimeters (i.e. with an along-track path resolution of 1 km). By considering the same ratio of measurements per distance crossed, the width of reservoir bodies observed by satellites like Sentinel-3 (i.e., resolution of 300 m) still needs to reach 3 km, which is larger than most reservoirs considered in our study – assuming that the satellite passed at the right location over them.

Then, for these reasons and as attested to by research studies that used these satellites, the focus has been made on reservoirs larger than 100 km$^2$ (Crétaux et al., 2016; Jiang et al., 2017) or rivers that stretch over several hundred kilometres (Villadsen et al., 2015).

**Proposed correction (p2, l11):** "They have a high vertical accuracy with root-mean-square errors on the order of centimetres to tens of centimetres depending on the altimeter and the size of the water body (Calmant et al., 2008; **Crétaux et al., 2016**). Yet, **the above-mentioned** sensors are affected by important drawbacks, including nadir viewing, narrow swath, coarse cross-track spacing (a few hundred kilometres), long along-track path length (about 1 km), large elevation differences around some water

areas, that impede their application to more than a few hundred large lakes and reservoirs on the planet (i.e. area > 100 km$^2$ and width > 500 m) (Crétaux and Birkett, 2006; Alsdorf et al., 2007; Gao et al., 2012). **More recent satellites such as Cryostat-2 or Sentinel-3 present significant improvements in terms of along-track resolution (≈300 m). However, their respective inter-track of 7 km and 52 km (Donlon et al., 2012; Crétaux et al., 2016) still place many reservoirs out of the trajectory of their nadir-viewing sensors onboard. The small inter-track of Cryosat is also realised at the expense of a long revisit cycle (369 days) that impedes any monitoring of small reservoirs on a monthly basis.**

**Proposed additional precision in the introduction (p1, l5):** "This paper proposes a novel approach using Landsat imagery and Digital Elevation Models (DEM) to retrieve information on storage variations in **any inaccessible region**."

**Proposed additional precision for the conclusion (p20, l15):** "For all "uncovered" small or large reservoirs, the uses of datasets available over the whole continental surface **make this method a valuable complement to satellite altimetry to increase the number of reservoirs observable anywhere in the world. The** thresholds dynamically defined for both the 2D enhancement and the 3D reconstruction **also** make the method potentially suitable to monitor reservoirs in **truly** inaccessible areas."

*Eqn (3): As pointed out in the short comment Eqn (3) does not make sense and yes NRMSE is a good solution.*

We thank you for this response to the first interactive comment. We consider your advice and keep the NRMSE criteria to quantify the error of our estimates.

*P 13, l8-10: "We can see coherent storage variations through the presence of drawdown-refill cycles, which means that the 2D enhancement and 3D reconstruction steps have improved the detection of water and helped to overcome the low Landsat repeat cycle of 16 days." I do not see this connection. How do you see that you improved the water detection when you are not comparing to anything?*

Yes indeed, this argument alone does not make sense. Thank you for noticing the mistake. Changes in water bodies' area obtained with similar tools – but without the 2D enhancement and 3D reconstruction – are available in the supporting information (Figure S3) of the study conducted by Müller et al. (2016). Improvements in the detection of annual drawdown-refill cycles are particularly clear for Sahwat al-Khadr and Roum dams.

**Proposed correction (p14, l6):** "**By qualitatively comparing our results to those obtained by Müller et al. (2016) (monitoring of Syrian reservoirs using Landsat 7 datasets but before the 2D and 3D corrections), we can see more** coherent storage variations through the presence of **annual** drawdown-refill cycles **– particularly for Roum and Sahwat al-Khadr. It** means that the 2D enhancement and 3D

reconstruction steps have improved the detection of water and helped to overcome the low Landsat repeat cycle of 16 days."

*P 14, l1-2. "Some of the differences between our estimates and measured data might then come from the inaccuracy regarding the data collection date". Is this because the in situ data are not daily?*

Yes indeed.

**Proposed correction (p14, l11):** "Reservoirs managed by Jordan are used to validate the method by comparing our remote sensing estimates of elevation and storage with **monthly** *in situ* measurements conducted by the Jordan Valley Authority (JVA)."

*Conclusion/Discussion: You could also mention the potential of Sentinel-1 and 2, which have a much higher resolution than Landsat.*

Yes indeed, thank you for mentioning these two satellites. Actually, as explained p2, l34, we chose to not detail SAR sensors as they have "been less used due to the difficulty to get consistent results, as the required condition of a significantly lower phase coherence of water areas than of the surrounding land surface is not always met with orbital repeat cycles of more than a few days, or with wind or rain (Alsdorf et al., 2007; Eilander et al., 2014)".

**Proposed correction (p2, l33):** "Water surface areas are commonly determined from optical satellite imagery such as MODerate Resolution Imaging Spectroradiometer (MODIS) and Landsat products (Xiao et al., 2006; Gao et al., 2012), or Synthetic Aperture Radar (SAR) sensors (e.g., RADARSAT, JERS-1, ERS or Sentinel-1) (Annor et al., 2009; Duan and Bastiaanssen, 2013; **Amitrano, 2014**)."

With regard to Sentinel 2, new references are added in the introduction.

**Proposed correction (p3, l8): "The potential of the recent two Sentinel-2 satellites can also be mentioned for post-2015 studies. Launched in June 2015 (Sentinel-2A) and March 2017 (Sentinel-2B), they provide spectral bands at a resolution of 10 m for visible and NIR bands, and at 20 m for SWIR bands. They also have a repeat cycle of 5 days by combining the two (European Space Agency, 2013; Yang et al., 2017)."**

**Proposed additional correction for the conclusion (p20, l22): "The recent two Sentinel-2 satellites also promise a great improvement of the method for post-2015 studies, as they produce images with spatial and temporal resolutions finer than Landsat (up to 10 m and 5 days). Combining Landsat and Sentinel-2 satellites would then reduce the already short revisit cycle of water bodies and would provide near real-time updates on water bodies storage.**

Furthermore, the algorithms used in the method automatically detect water bodies, define the water areas retrieval parameters, build filling curves and assess reservoir storage. **Such algorithmic tools can**

then be dynamically updated with each new image from Sentinel-2 and Landsat satellites, giving the model the potential to learn by itself and correct previous storage estimates while generating new ones. This approach is somehow comparable to the continuous change detection proposed by Zhu and Woodcock (2014)."

**References added or not cited in the first version of the discussion paper:**

Amitrano, D., Martino, G. D., Iodice, A., Mitidieri, F., Papa, M. N., Riccio, D., and Ruello, G.: Sentinel-1 for Monitoring Reservoirs: A Performance Analysis, Remote Sensing, 6, 10 676 – 10 693, doi:10:3390/rs61110676, 2014.

Crétaux, J.-F., Abarca-del Río, R., Bergé-Nguyen, M., Arsen, A., Drolon, V., Clos, G., and Maisongrande, P.: Lake Volume Monitoring from Space, Surveys in Geophysics, 37, 269 – 305, doi:10:1007/s10712-016-9362-6, 2016.

Donlon, C., Berruti, B., Buongiorno, A., Ferreira, M.-H., Féménias, P., Frerick, J., Goryl, P., Klein, U., Laur, H., Mavrocordatos, C., Nieke, J., Rebhan, H., Seitz, B., Stroede, J., and Sciarra, R.: The Global Monitoring for Environment and Security (GMES) Sentinel-3 mission, Remote Sensing of Environment, 120, 37 – 57, doi:10:1016/j:rse:2011:07:024, 2012.

European Space Agency: CryoSat Product Handbook, available at: https://earth.esa.int/documents/10174/125272/CryoSat_Product_Handbook, 2012.

European Space Agency: Sentinel-2 Mission Details, available at: https://earth.esa.int/web/guest/missions/esa-operational-eo-missions/sentinel-2, 2013.

Jiang, L., Nielsen, K., Andersen, O. B., and Bauer-Gottwein, P.: Monitoring recent lake level variations on the Tibetan Plateau using CryoSat-2 SARIn mode data, Journal of Hydrology, 544, 109 – 124, doi:10:1016/j:jhydrol:2016:11:024, 2017.

Villadsen, H., Andersen, O. B., Stenseng, L., Nielsen, K., and Knudsen, P.: CryoSat-2 altimetry for river level monitoring – Evaluation in the Ganges-Brahmaputra River basin, Remote Sensing of Environment, 168, 80 – 89, doi:10:1016/j:rse:2015:05:025, 2015.

Yang, X., Zhao, S., Qin, X., Zhao, N., and Liang, L.: Mapping of Urban Surface Water Bodies from Sentinel-2 MSI Imagery at 10 m Resolution via NDWI-Based Image Sharpening, Remote Sensing, 9, doi:10:3390/rs9060596, 2017.

Zhu, Z. and Woodcock, C. E.: Continuous change detection and classification of land cover using all available Landsat data, Remote Sensing of Environment, 144, 152 – 171, doi:/10:1016/j:rse:2014:01:011, 2014.

**Response to the Referee Comment 2 posted by Webster Gumindoga (Referee 2):**

*I read with enthusiasm the paper by Nicolas Avisse et al on Monitoring small reservoirs storage from satellite remote sensing in inaccessible areas. The approach to use satellite data (Landsat imagery and Digital Elevation Models (DEM)) to retrieve information on storage variations in ungauged and inaccessible areas is welcome for improved water resource management.*

Thank you for your interest and for taking time commenting our paper.

*A question arises for the Fmask function for distinguishing land and water areas and producing a probability mask for clouds. What specific criteria was used to manually remove images that are almost entirely covered by clouds or with obvious large errors in water bodies detection? What specific quality control measures did the authors take to remain with 245 images per location? The authors can do justice by quantifying the uncertainty in the Fmask method.*

The analysis conducted to "manually remove images that are almost entirely covered by clouds or with obvious large errors in water bodies detection" (p6, l3) is a rough observation of Fmask classification results (mainly for categories 'clouds' and 'water' as mentioned above). The quality control is a visual comparison between these classification results and original images (SWIR-R-G for instance). Zhu et al. (2012) evaluate a cloud overall accuracy of 96.41 %, but it depends a lot on the satellite, location and time: Zhu et al. (2015) estimate an overall accuracy (i.e., for all classes) varying between 24 % and 89 %, for instance, depending on the Landsat 8 image chosen. Also, according to our study (p6, l8): "on average, 24.1 % of reservoirs' pixels are misclassified as land, 8.1 % are covered with clouds or cloud shadows, and 8.6 % are in 'N/A' areas".

In fact, the objective of this step is not to precisely detect clouds or water areas. We just need a first rough selection of images and remove those that could affect the next statistical analyses (statistical correction of elevation and 3D reconstruction through hidden areas). For instance, if Fmask detects clouds over the whole image, then it cannot be used in the next steps. Similarly, if Fmask classifies half an image as water, it is obviously a misdetection from the algorithm. By removing such images, we went from 300 to 245 images per location.

**Proposed correction (p5, l16):** "The algorithm was originally designed to separate potential cloud pixels from clear sky pixels on Landsat images using empirical thresholds on NDVI and the near-infrared band, **with an overall accuracy of 96.41 % (Zhu and Woodcock, 2012)**."

*In Section 2.1.3, how realistic is to define automatically the threshold for optimally distinguishing water bodies from clouds using the MNDWI technique?*

As the referee W. Gumindoga rightly points out, an automatic classification with MNDWI or NDVI (or any other criterion) will not give as good results as if we chose a specific criterion for each reservoir at each

time. A trade-off is indeed required between the time to spend on the detection and the quality of the results.

As explained in the introduction (p3, l11-24), various methods have been applied to detect water areas. The most basic ones rely on a predefined NDVI or MNDWI threshold, which is problematic for a multi-temporal analysis (Liu et al., 2012). Coltin et al. (2016) give an inventory of other indices generally used for detecting water, and advocate the implementation of automatic thresholds as they develop a supervised learning approach. Other methods rely on an automatic unsupervised classification (Wang et al., 2008; Gao et al., 2012). In our paper, we choose to automatically define a threshold for each image. Our protocol has actually the advantage of being entirely automatic (no further association between class and type of land use for instance). This approach is very fast, no selection of reservoir approximate location is required, and, as mentioned in the conclusion, it could "provide near real-time updates on water bodies storage".

**Proposed additional reference (p3, l13):** "But determining an adequate value for a multi-temporal analysis can be challenging because such a threshold is known to be case-dependent (Liu et al., 2012; **Coltin et al., 2016**)."

**Other additional correction (p3, l17):** "To address these issues, decision tree defined thresholds have successfully been applied with various vegetation indices (e.g., Xiao et al., 2006; Islam et al., 2010; Yan et al., 2010), but remain case-dependent. **Coltin et al. (2016) have then advocated the implementation of automatic thresholds as they developed a supervised learning approach to improve flood mapping.**"

*Authors can also justify the selection of Landsat 7 images over the more recent Landsat 8, which do not have stripes after all.*

Yes you are right, Landsat 8 do have the advantage of not having stripes. We actually used all kinds of Landsat images including Landsat 8: "about 300 Landsat 4, 5, 7 and 8 images for each scene […] are downloaded from the [USGS website]" (p5, l10). The goal is to use all Landsat images available to analyse changes in reservoir storage over long periods of time (ideally several decades), and Landsat 8 images are only available from February 2013.

*Section 3.1 what do the authors mean by saying "...some of the differences between our estimates and measured data might then come from the inaccuracy regarding the data collection date."*

As pointed out by the Anonymous Referee 1, we did not mention that the "*in situ* measurements conducted by the Jordan Valley Authority" are monthly. Then, as these measurements are not automatically recorded, we do not know exactly on which day they were collected, and if they are always collected the same day in the month. Such uncertainty may change the difference between our estimates and measured data.

**We proposed the following correction when answering the Referee 1 comment (p14, l11):** "Reservoirs managed by Jordan are used to validate the method by comparing our remote sensing estimates of elevation and storage with **monthly** *in situ* measurements conducted by the Jordan Valley Authority (JVA)."

**Proposed additional correction (p15, l4):** "**Because no information is available regarding the data collection date,** some of the differences between our estimates and measured data might then come from **this lack of metadata.**"

*The authors need to improve on the equality of the maps by improving on some map fundamentals/basics such as north arrow, legend and scale.*

As pointed out by T. Francke in the Short Comment 1, we indeed forgot to specify that 1 unit equals 1 m for the scale.

**We have updated the legend of figures 1, 3, 4, 7 and 9. We have also added "latitude" and "longitude" for axis titles to match common representations of satellite images. The title "Inundation frequency" has also been added next to the colorbar in Figure 3.**

*Why not validating the elevation-area relationships with some established/measured rating curves*

We unfortunately do not have such relationships for Syrian nor Jordanian reservoirs. As the observed elevation and volume for Jordanian reservoirs do not represent the whole range of possibilities, and because few elevation measurements were available ($15 \leq N \leq 35$, see p17, l14) and not necessarily conducted at the same time as storage measurements, the relationships could not be retrieved with precision.

**New reference:**

[revised manuscript text omitted]